# Failure Prediction at Runtime for Generative Robot Policies

**Ralf Römer**[1,*]    **Adrian Kobras**[1,*]    **Luca Worbis**[1]    **Angela P. Schoellig**[1,2,3]

[1] Technical University of Munich, Germany; Learning Systems and Robotics Lab;
Munich Institute of Robotics and Machine Intelligence (MIRMI)
[2] Robotics Institute Germany    [3] Munich Center for Machine Learning

ralf.roemer@tum.de

## Abstract

Imitation learning (IL) with generative models, such as diffusion and flow matching, has enabled robots to perform complex, long-horizon tasks. However, distribution shifts from unseen environments or compounding action errors can still cause unpredictable and unsafe behavior, leading to task failure. Therefore, early failure prediction during runtime is essential for deploying robots in human-centered and safety-critical environments. We propose FIPER, a general framework for Failure Prediction at Runtime for generative IL policies that does not require failure data. FIPER identifies two key indicators of impending failure: (i) out-of-distribution (OOD) observations detected via random network distillation in the policy's embedding space, and (ii) high uncertainty in generated actions measured by a novel action-chunk entropy score. Both failure prediction scores are calibrated using a small set of successful rollouts via conformal prediction. A failure alarm is triggered when both indicators, aggregated over short time windows, exceed their thresholds. We evaluate FIPER across five simulation and real-world environments involving diverse failure modes. Our results demonstrate that FIPER better distinguishes actual failures from benign OOD situations and predicts failures more accurately and earlier than existing methods. We thus consider this work an important step towards more interpretable and safer generative robot policies. Code, data, and videos are available at tum-lsy.github.io/fiper_website.

## 1   Introduction

Imitation learning (IL) for robotics usually involves high-dimensional, diverse, and multimodal data [69, 39]. Recent advances in generative modeling, such as diffusion models [64, 25] and flow matching [40], have led to significant progress in IL algorithms, substantially expanding the range of complex, long-horizon tasks that robots can perform [11, 78, 56, 6, 5]. Despite these improvements, generative IL policies remain imperfect even when trained on large-scale robot data [39, 50]: Unexpected visual or state shifts and compounding errors in action predictions can cause erratic or unsafe behavior, ultimately leading to task failure [37, 1, 42, 75]. In human-centered and safety-critical settings, it is therefore crucial to predict such failures as early as possible during runtime to enable timely intervention [44] or safe fallbacks [8] or to ask human experts to demonstrate the task [74]. However, failure prediction is difficult because it cannot be treated as a typical classification problem for two main reasons: First, it is often not possible to generate examples of failures [18, 42], as this would endanger the robot and its environment. Second, the

---

*Equal contribution.

39th Conference on Neural Information Processing Systems (NeurIPS 2025).

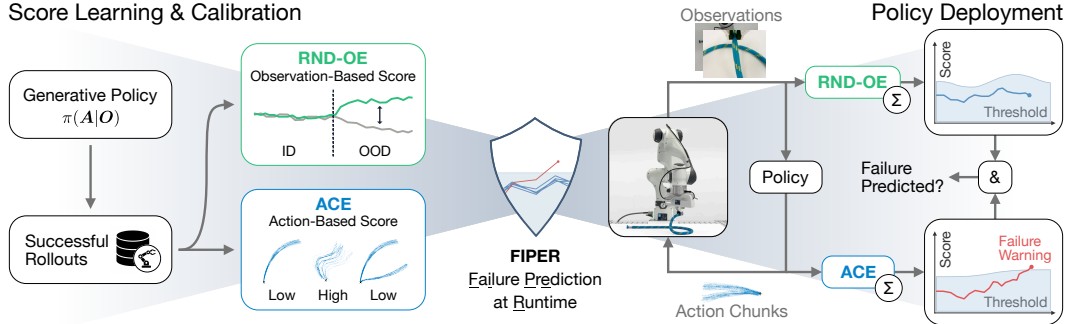

Figure 1: FIPER can predict task failures of generative robot policies during runtime without using any failure data. FIPER detects two key signals indicative of impending failure: (i) consecutive out-of-distribution (OOD) observations via random network distillation in the policy's observation embedding space (RND-OE) and (ii) persistently high uncertainty in generated actions via action-chunk entropy (ACE). Both scores are calibrated based on a few successful rollouts and aggregated over a sliding window. If both submodules issue a warning, then FIPER predicts a failure. Its task-agnostic design enables FIPER to issue accurate and early failure warnings in diverse environments.

range of potential failure modes during closed-loop operation in complex, real-world environments is vast [1, 75], meaning that creating labeled data is not a viable option.

Existing approaches for anticipating and detecting policy failures fall into two broad categories: Those that look for out-of-distribution (OOD) observations or actions [37, 75, 22] and those that externally monitor the robot's behavior or progress using vision-language-models (VLMs) [1, 17]. Both approaches have drawbacks: Pure OOD detectors trigger on any novel situation, even if the policy can generalize to it. At the same time, VLM-based methods only raise alarms after errors manifest, providing no foresight about impending failure. Most closely related to our work, Xu et al. [75] propose using a scalar uncertainty score derived from the policy inputs. However, as future behavior is determined by the *actions* of the policy, observation-only methods can miss early warning signs present in the action distribution. In summary, current approaches either isolate observations and actions, depend on failure-mode labels, or issue alerts too late for intervention.

To address these limitations, we propose Failure Prediction at Runtime (FIPER), a lightweight framework for predicting failures of generative robot policies that requires no examples of failures. FIPER is built on our insight that actual failures coincide with two indicators: (i) successive deviations of the observations from the patterns expected in successful rollouts and (ii) prolonged high uncertainty in the stochastic actions generated by the policy. We propose measuring (i) by leveraging random network distillation (RND) [9] in the policy's observation embedding space and (ii) based on the entropy in the observation-conditioned action chunk distributions. We statistically calibrate our observation- and action-based uncertainty scores on a small set of successful rollouts using conformal prediction [3]. At runtime, FIPER raises an alarm when *both* scores, aggregated over short time windows, exceed their respective thresholds. This design, illustrated in Figure 1, enables FIPER to predict failures accurately *and* at an early stage with a low rate of false alarms. We evaluate FIPER for widespread diffusion- and flow-based IL policies across diverse simulation and real-world environments that exhibit various types of failures. In comparison to state-of-the-art baselines, our proposed scores can better distinguish actual failures from OOD situations, and FIPER achieves the highest accuracy and lowest detection time across all environments. Our findings suggest that FIPER is a promising step towards more interpretable and safer deployment of generative robot policies.

## 2 Related Work

**Generative Policies for IL:** IL is an effective method to teach a robot new tasks by training a policy directly on expert demonstrations [55, 77, 45], but has historically suffered from compounding errors due to (covariate) distribution shifts [58, 59] as well as limited ability to capture multimodal expert behavior [38, 31]. Recent progress in IL [78, 79, 28, 2, 52, 11] can be mainly attributed to the development of powerful generative modeling techniques [34, 64, 65, 40] and the availability of high-quality demonstration data [32, 50, 10]. Generative diffusion models [64, 25, 65] in particular

excel at capturing the high-dimensional, multimodal distributions commonly encountered in human demonstration data [69]. Diffusion Policy [11] first demonstrated the potential of using a diffusion model conditioned on visual observations for receding horizon control in complex manipulation tasks. Further studies have extended this approach to additional input modalities like language or touch [56, 24], improved robustness [71], or incorporated multi-task capabilities [72]. Recently, generative models using flow matching [40] instead of diffusion have demonstrated faster inference and improved trajectory smoothness for IL policies [6, 7, 26, 5]. To improve generalization, recent work has also trained IL policies on large-scale, cross-embodiment data [50], with the option to fine-tune them for a particular robot or task [33, 68, 6, 43]. Despite the impressive capabilities of generative IL policies, they inherit the fundamental properties of IL, often behaving unpredictably and failing when deployed beyond their training distribution [35].

**Uncertainty Quantification and OOD Detection:**   Detecting OOD samples is a key challenge across various subfields of machine learning [76, 60, 63], with numerous methods proposed to address it. Some approaches rely on a priori access to OOD data [23, 16, 42, 70], which is, however, often unavailable or not comprehensive enough [53]. Another line of work directly measures epistemic uncertainty [27], using model calibration [20], ensembling methods [36, 49, 46, 61] or Monte-Carlo dropout (MC) [19, 49, 13], which can then be used to detect OOD cases. Random network distillation (RND) [9], originally proposed to incentivize exploration in reinforcement learning (RL), has been shown to outperform both deep ensembles and MC dropout [12]. Prior work has adopted RND to uncertainty quantification [12] and confidence estimation in offline RL [22]. Other approaches use the reconstruction error or latent embeddings of autoencoders [57, 73, 30, 47, 51] or directly measure deviations in the observation or embedding space [67, 62, 51] to detect dissimilarity from the training distribution. Our method combines RND in the observation embedding space with a novel entropy-based score, which is conceptually related to Bayesian methods [19] and to the idea that sampling a batch instead of a single action can be used to detect failures [1].

**Failure Detection for IL Policies:**   Reliable handling and detection of policy failures is crucial to ensure safe deployment of robots, especially in human-centered environments [54, 42]. OOD detection alone is insufficient for recognizing failures in IL, as the policy might be able to generalize in some novel situations. The failure prediction problem is also complicated by the closed-loop operation and multimodal behavior of generative policies, which can generate very different actions for the same observation [1]. Agia et al. [1] propose addressing this challenge by combining a cumulative temporal consistency score that quantifies the divergence between the overlapping components of two consecutive action distributions, with a VLM to detect task progression failures. Using VLMs for monitoring policies and reasoning about failures has also been proposed in other works [17], but these approaches are inherently slow and unable to predict failures early. Liu et al. [42] train a failure detector on image embeddings of a visual world model and identify OOD states using clustering. Still, their method relies on failure examples similar to other works [18]. ReDiffuser [22] uses an RND model trained on trajectories to estimate the reliability of sampled decisions and select the most reliable one. However, ReDiffuser is tailored to state-based policies and cannot account for high-dimensional visual inputs. Lee et al. [37] use the loss of a diffusion policy to predict failures, but their approach is not directly applicable to other generative models, such as flow matching. FAIL-Detect [75] fits a flow matching model to the distribution of observation embeddings and detects policy failures by quantifying the likelihood of observations under the learned distribution. Existing methods [1, 75, 42, 22] have in common that they aim to detect failures either only from the policy inputs *or* outputs. In contrast, FIPER takes both into account, allowing us to better distinguish actual failures from benign OOD situations that the policy can handle.

## 3   Problem Setup

We consider a robotic system performing a given task, which we model as a Markov decision process (MDP) with observation $\boldsymbol{o}_k \in \mathcal{O}$ and action $\boldsymbol{a}_k \in \mathcal{A}$ at timestep $k = 0, 1, \ldots$. We assume a finite horizon $T$ for the MDP, but the task can also be completed in fewer than $T$ timesteps. A generative IL policy $\pi(\boldsymbol{A}|\boldsymbol{O})$ is trained to match the observation-conditioned action distribution in a demonstration dataset $\mathcal{D}$. At each policy timestep $t = 0, h, 2h, \ldots$, the learned policy generates a chunk (sequence) of $H$ future actions $\boldsymbol{A}_t = (\boldsymbol{a}_{t|t}, \ldots, \boldsymbol{a}_{t+H-1|t}) \sim \pi(\cdot|\boldsymbol{O}_t)$ conditioned on an observation history $\boldsymbol{O}_t = (\boldsymbol{o}_{t-T_{\mathrm{h}}+1}, \ldots, \boldsymbol{o}_t)$ of length $T_{\mathrm{h}} \geq 1$. The first $h \leq H$ actions $\boldsymbol{a}_{t:t+h-1|t}$ are applied

to the system before re-planning at timestep $t + h$. This general policy formulation encompasses common, state-of-the-art IL methods [11, 78] and vision-language-action models (VLAs) [6, 33].

Depending on the initial condition $o_0$, the stochastic transitions, and the stochastic actions generated by the policy, the system may succeed or fail to complete the task within $T$ timesteps. Our goal is to detect as *accurately* and *early* as possible situations for which the policy, when executed for the remaining timesteps up to $T$, would not succeed in completing the task. For this, we aim to design a failure predictor $F(\cdot)$ that takes the trajectory $\boldsymbol{\tau}_{:t} = (\boldsymbol{O}_0, \boldsymbol{A}_0, \boldsymbol{O}_h, \boldsymbol{A}_h, \ldots, \boldsymbol{O}_t, \boldsymbol{A}_t)$ up to the current timestep $t$ as input and predicts the final rollout outcome $F(\boldsymbol{\tau}_{:t}) \in \{0, 1\}$. If at any time $t$, $F(\boldsymbol{\tau}_{:t}) = 1$, the rollout is flagged as Fail with detection time $t$.

## 4 Methodology

We design FIPER to detect two characteristics of policy failures: Consecutive OOD observations and high uncertainty in the conditional action distributions. These failure indications are handled by two submodules, whose outputs are combined in the overall failure predictor $F(\cdot)$. We do not rely on failure data and use only a few successful ID policy rollouts for training and calibration.

### 4.1 Detecting OOD Observations via Random Network Distillation (RND-OE)

Generative IL policies have demonstrated some generalization capabilities [11, 78], but they still often struggle in situations that deviate substantially from their training distribution [39]. To predict policy failures based on observations, we therefore aim to detect if $\boldsymbol{O}_t$ differs from ID situations in a way that is likely to negatively affect the policy's performance. For this purpose,

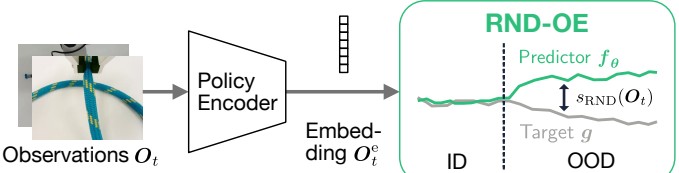

Figure 2: RND-OE recognizes failure-prone out-of-distribution (OOD) situations using random network distillation (RND) in the policy's observation embedding space.

we adopt RND [9], a novelty detector first proposed for exploration in reinforcement learning. RND consists of two neural networks with $m$-dimensional outputs, a randomly initialized target network $\boldsymbol{g}(\cdot)$ and a predictor network $\boldsymbol{f}_{\boldsymbol{\theta}}(\cdot)$ parameterized by $\boldsymbol{\theta}$. The target network is frozen, and the predictor network is trained to match the outputs of the target network on ID data. We apply RND to the policy inputs and train the predictor on ID success data $\mathcal{D}_{\text{ID}} \sim q_{\pi}$ to minimize the loss

$$\mathcal{L}(\boldsymbol{\theta}) = \mathbb{E}_{(\boldsymbol{O}_t, \boldsymbol{A}_t) \sim \mathcal{D}_{\text{ID}}} \big[ s_{\text{RND}}(\boldsymbol{O}_t) \big], \qquad \text{where} \qquad s_{\text{RND}}(\boldsymbol{O}_t) = \|\boldsymbol{f}_{\boldsymbol{\theta}}(\boldsymbol{O}_t) - \boldsymbol{g}(\boldsymbol{O}_t)\|_2. \quad (1)$$

Intuitively, the predictor and target networks produce similar outputs for observations they have seen before, but their outputs diverge on novel, unseen data. We reuse and freeze the observation encoder $\boldsymbol{h}(\cdot)$ of the policy in both networks, which has two benefits: First, we can detect anomalies directly in the policy's embedding space, which are more indicative of failures than OOD raw observations. Second, using the pre-trained feature extractor $\boldsymbol{h}(\cdot)$ allows us to train the RND model even from a small dataset $\mathcal{D}_{\text{ID}}$. Due to this design, we refer to $s_{\text{RND}}(\boldsymbol{O}_t)$ as RND observation embedding (RND-OE) score. To ensure the predictor $\boldsymbol{f}_{\boldsymbol{\theta}}$ is expressive enough to approximate the target network $\boldsymbol{g}$ well on the ID dataset, we design $\boldsymbol{f}_{\boldsymbol{\theta}}$ to be slightly larger than $\boldsymbol{g}$ (see Appendix A.2). After training, we can use $s_{\text{RND}}(\boldsymbol{O}_t)$ to measure how much an observation embedding $\boldsymbol{O}_t^{\text{e}} = \boldsymbol{h}(\boldsymbol{O}_t)$, which action generation is effectively conditioned on, deviates from the patterns in successful ID rollouts. This idea is visualized in Figure 2.

Recent IL policies [11, 78] can often handle brief and less severe OOD situations, especially when trained on massive data [39]. However, multiple consecutive OOD observations are likely to cause compounding errors in action predictions [4] from which the policy cannot recover. For this reason, we construct our observation-based failure prediction score $\eta_O(\boldsymbol{\tau}_{:t})$ by aggregating the RND-OE score $s_{\text{RND}}(\boldsymbol{O}_t)$ over a sliding time window of size $w_O$ as

$$\eta_O(\boldsymbol{\tau}_{:t}) = \sum_{k=0}^{\min(w_O - 1, t/h)} s_{\text{RND}}(\boldsymbol{O}_{t-kh}) = \underbrace{s_{\text{RND}}(\boldsymbol{O}_t) + s_{\text{RND}}(\boldsymbol{O}_{t-h}) + \ldots}_{\leq w_O \text{ times}}. \quad (2)$$

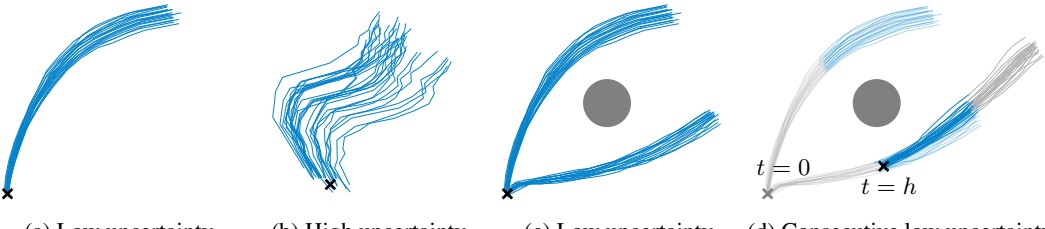

(a) Low uncertainty    (b) High uncertainty    (c) Low uncertainty    (d) Consecutive low uncertainty

Figure 3: (a) to (c) In imitation learning from multimodal demonstrations, uncertainty in generated actions is reflected in entropy rather than variance. (d) Our action-chunk entropy (ACE) score is designed to handle observation-dependent action multimodality, whereas STAC [1] typically associates timesteps at which the policy decides on a behavior mode with high uncertainty.

If $\eta_O(\boldsymbol{\tau}_{:t})$ exceeds a threshold $\gamma_{O,t}$ at any policy timestep $t$, the recent observations indicate imminent task failure, i.e., our observation-based failure predictor is given by

$$F_O(\boldsymbol{\tau}_{:t}) = \mathbb{1}(\eta_O(\boldsymbol{\tau}_{:t}) > \gamma_{O,t}). \tag{3}$$

We will discuss the calculation and calibration of the threshold $\gamma_{O,t}$ in Section 4.3.

## 4.2 Detecting High Action Uncertainty via Action-Chunk Entropy (ACE)

While we can infer *current* OOD situations from the observations, the *future* system evolution is ultimately determined by the actions that are generated by the policy and applied to the system. To improve model interpretability, which is especially important in decision-making systems [20], we aim to capture uncertainty at an intent level. To this end, we leverage the conditional action distribution $\pi(\boldsymbol{A}|\boldsymbol{O}_t) = \pi(\boldsymbol{A}|\boldsymbol{O} = \boldsymbol{O}_t)$ at the current observation $\boldsymbol{O}_t$ to predict failures. Data for IL often contains action multimodality, with the number of modes generally being observation-dependent and unknown [29, 69]. Thus, a generative IL policy can generate very different (e.g., in an $L_2$ sense) actions for the same observation in a successful rollout. For this reason, merely measuring the variance in generated actions tells us little about uncertainty, as illustrated in Figure 3 (a) to (c). However, we observe that action multimodality in IL is usually of a discrete nature. For example, a robot can first pick up object A, B *or* C, grasp objects from the side *or* above, and avoid obstacles left *or* right. Therefore, each generated action should clearly correspond to one of the observation-dependent modes for successful task completion.

In principle, sharpness of the modes of $\pi(\boldsymbol{A}|\boldsymbol{O}_t)$ can be quantified by the entropy $E_{\boldsymbol{O}_t}(\boldsymbol{A}) = -\int_{\mathcal{A}^H} p_\pi(\boldsymbol{A}_t|\boldsymbol{O}_t)\log p_\pi(\boldsymbol{A}_t|\boldsymbol{O}_t)\mathrm{d}\boldsymbol{A}_t$. However, the likelihood $p_\pi(\boldsymbol{A}_t|\boldsymbol{O}_t)$ is unknown for diffusion- or flow-based policies $\pi$, so we need to approximate $E_{\boldsymbol{O}_t}(\boldsymbol{A})$. For this, we sample a batch $\mathbf{A}_t = \left(\boldsymbol{A}_t^{(1)}, \ldots, \boldsymbol{A}_t^{(B)}\right)$ of $B$ action chunks $\boldsymbol{A}_t^{(j)} = \left(\boldsymbol{a}_{t|t}^{(j)}, \ldots, \boldsymbol{a}_{t+H-1|t}^{(j)}\right) \sim \pi(\cdot|\boldsymbol{O}_t)$, $j = 1, \ldots, B$, at each policy timestep $t$. Since the dimensionality of $\boldsymbol{A}_t \in \mathcal{A}^H$ grows exponentially with the action chunk length $H$, directly approximating $E_{\boldsymbol{O}_t}(\boldsymbol{A})$ would require an excessively large batch size $B$, exceeding the parallelization capabilities of common GPUs. As we aim for failure prediction *during runtime*, we reduce computational complexity by treating the prediction timesteps $t$ through $t + H - 1$ separately and define the action-chunk entropy (ACE) score as

$$s_{\mathrm{ACE}}(\mathbf{A}_t) = \sum_{i=0}^{H-1} \hat{E}\left(\boldsymbol{a}_{t+i|t}^{(1)}, \ldots, \boldsymbol{a}_{t+i|t}^{(B)}\right). \tag{4}$$

Here, $\hat{E}(\cdot)$ measures the entropy of an action prediction step $t+i$ based on the actions sampled for that timestep. We adopt a dimension-wise binning approach to calculate $\hat{E}(\cdot)$, which we found to be more computationally efficient, robust, and easier to tune than other methods [14, 1] (see Appendix A.1 for details). Similar to the observation-based score (2), we aggregate (4) over a sliding window of $w_A$ policy timesteps and define our action-based failure predictor $F_A(\cdot)$ with a threshold $\gamma_{A,t}$, i.e.,

$$\eta_A(\boldsymbol{\tau}_{:t}) = \sum_{k=0}^{\min\{w_A-1, t/h\}} s_{\mathrm{ACE}}(\mathbf{A}_{t-kh}) = \underbrace{s_{\mathrm{ACE}}(\mathbf{A}_t) + s_{\mathrm{ACE}}(\mathbf{A}_{t-h}) + \ldots}_{\leq w_A \text{ times}}, \tag{5}$$

$$F_A(\boldsymbol{\tau}_{:t}) = \mathbb{1}(\eta_A(\boldsymbol{\tau}_{:t}) > \gamma_{A,t}). \tag{6}$$

### 4.3 Observation- AND Action-Based Failure Prediction with FIPER

Not all OOD observations lead to failure, and there may be temporary high (aleatoric) uncertainty in the generated actions even in successful rollouts, for example, due to the suboptimality and diversity of demonstrations. To obtain robust predictions specifically about *task failure*, we flag a rollout as `Fail` if and only if both failure predictors (3) and (6) raise a warning, i.e., FIPER combines their outputs with a logical conjunction as

$$F(\boldsymbol{\tau}_{:t}) = F_O(\boldsymbol{\tau}_{:t}) \wedge F_A(\boldsymbol{\tau}_{:t}) = \mathbb{1}(\eta_O(\boldsymbol{\tau}_{:t}) > \gamma_{O,t} \wedge \eta_A(\boldsymbol{\tau}_{:t}) > \gamma_{A,t}). \quad (7)$$

To calibrate the two thresholds $\gamma_{O,t}$ and $\gamma_{A,t}$ in (7), we can leverage conformal prediction (CP) [3] for functional data [15]. For this, we use a calibration dataset $\mathcal{D}_c = \{\boldsymbol{\tau}^i\}_{i=1}^M \sim q_\pi$ of $M$ i.i.d. successful policy rollouts $\boldsymbol{\tau}^i = (\boldsymbol{O}_0^i, \boldsymbol{A}_0^i, \boldsymbol{O}_h^i, \boldsymbol{A}_h^i, \ldots, \boldsymbol{O}_{T_i}^i, \boldsymbol{A}_{T_i}^i)$. Since the uncertainty scores (2) and (5) vary considerably during each rollout, often being smallest at $t = 0$, we design their respective thresholds to be time-varying. We compute $\gamma_{O,t}$ and $\gamma_{A,t}$ in a similar way and, therefore, focus on the former here for brevity. As proposed by Diquigiovanni et al. [15], we split $\mathcal{D}_c$ into two disjoint parts and use them to separately calculate the time-varying mean $\mu_{O,t}$ and band-width $b_{O,t}(\delta)$ of the score signals $\eta_O(\boldsymbol{\tau}_{:t}^i)$, $t = 0, h, \ldots, i = 1, \ldots, M$. Thereby, $1 - \delta$ controls the proportion of signals staying in $[\mu_{O,t} - b_{O,t}(\delta), \mu_{O,t} + b_{O,t}(\delta)]$ for the entire time (see Appendix A.3 for more details). Since $\eta_O \geq 0$ by design, we only require an upper time-varying threshold $\gamma_{O,t} = \mu_{O,t} + b_{O,t}$ to enforce a desired upper bound on the probability of raising false alarms with (3) (similarly for (6)). As $\eta_O(\boldsymbol{\tau}_{:t})$ and $\eta_A(\boldsymbol{\tau}_{:t})$ are not independent, our combined predictor (7) satisfies the same bound.

**Proposition 1.** *Set $\delta \in (0,1)$, and define the thresholds $\gamma_{O,t}$ and $\gamma_{A,t}$ as described above. Then, the probability that the failure predictor (7) of* FIPER *flags a new successful ID rollout $\boldsymbol{\tau} \sim q_\pi$ of length $T' \leq T$ as* `Fail` *at any policy timestep $t \leq T'$ satisfies the upper bound*

$$P(\exists t \in \{0, h, \ldots, T'\} \text{ s.t. } F(\boldsymbol{\tau}_{:t}) = 1) \leq \delta. \quad (8)$$

A more rigorous formulation of Proposition 1 and a proof are given in Appendix A.4. We can view $\delta$ as a design parameter, with a larger value increasing sensitivity to failures at the expense of more false alarms. Proposition 1 quantifies the ability of FIPER to recognize successes as such, not failures. To derive such a result, we would have to assume the availability of failure data, which we explicitly do not do for the reasons above. In addition to the threshold definition above, which we refer to as one-sided *CP band*, we also investigate two alternatives: A *CP constant* threshold $\gamma_{O,t} = \gamma_O$ defined as the $1 - \delta$ quantile of the set of conformity scores $\{\max_t \eta_O(\boldsymbol{\tau}_{:t}^i)\}_{i=1}^M$ and a simpler *time-varying* threshold $\gamma_{O,t}$ defined as the $1 - \delta$ quantile of $\{\eta_O(\boldsymbol{\tau}_{:t}^i)\}_{i=1}^M$. Proposition 1 also applies to the CP constant but not to the time-varying threshold.

*Remark* 1 (Rollout data). The training data for RND-OE, $\mathcal{D}_{ID}$, and the calibration data $\mathcal{D}_c$ are sampled from the same distribution $q_\pi$ but would need to be disjoint for Proposition 1 to hold rigorously. However, setting $\mathcal{D}_{ID} = \mathcal{D}_c$ worked well in our experiments, so we collect only one rollout dataset.

## 5 Experiments

We conduct extensive experiments across a diverse set of environments, mainly aimed at answering the following research questions: **Q1)** How well can our proposed scores distinguish failures from mere OOD situations? **Q2)** Does FIPER benefit from combining an observation- and an action-based failure predictor? **Q3)** Can FIPER predict failures more accurately and earlier than existing methods?

**Environments.** We consider three popular benchmark environments, SORTING [29], STACKING [29] and PUSHT [11], and two real-world tasks, PRETZEL and PUSHCHAIR [1]. These environments differ in terms of robot embodiment, observation and action space, degree of action multimodality, and task duration, thus representing a diverse set of failure modes. We briefly explain the environments and how we create OOD scenarios to induce policy failures in the following:

- SORTING: A Franka robot needs to push two blocks on a table into their color-matching boxes. OOD: We change the dimensions of the blocks and the positions of the target boxes.
- STACKING: A Franka robot needs to stack three blocks on a target area. The task is multimodal, as there are six possible block arrangements. OOD: We vary the block sizes and target location.
- PUSHT: The agent must push a planar T-shaped object into a target configuration. OOD: We use the data from Sentinel [1], which includes variations in the shape and dimensions of the T-object.

- PRETZEL: A Franka robot needs to fold a rope into a pretzel-like shape. OOD: We vary the rope's initial configuration and rotate it around its own axis, which changes the bending behavior.
- PUSHCHAIR: A mobile manipulator needs to push a chair to a target position. OOD: We use the data from Sentinel [1], which includes variations of the initial chair pose.

**Implementation.** Our IL policies take one or two RGB images and the robot's proprioceptive information as inputs. We consider different generative modeling techniques and policy backbones: For PUSHT, PRETZEL, and PUSHCHAIR, we use denoising diffusion [25] with a temporal U-Net [11] backbone, and for SORTING and STACKING, we use flow matching [40] with a transformer backbone from ACT [78]. As image encoder, we use ResNet-18 [21] due to its demonstrated effectiveness for feature extraction in robotics [11, 80, 29]. We compute the ACE score in the Cartesian space of predicted end-effector positions to obtain task-relevant and interpretable uncertainty information in a computationally efficient way. For RND-OE, we use an MLP with 6 (4) layers for the predictor (target) network. Further details on model architectures, hyperparameters, and training are provided in Appendix B.

**Baselines.** We consider four state-of-the-art baselines for recognizing failures of IL policies. For all methods that involve offline training or clustering, we use the same ID success data $\mathcal{D}_{\mathrm{ID}}$.

- PCA-kmeans [42] computes and clusters the principal components of the observation embeddings in $\mathcal{D}_{\mathrm{ID}}$. During runtime, the distance of the same principal components of $\boldsymbol{O}_t^{\mathrm{e}}$ to the nearest cluster center is calculated to detect OOD observations.
- logpZO [75] learns the distribution of observation embeddings via flow matching. For a new observation, starting from $\boldsymbol{O}_t^{\mathrm{e}}$, the ODE is solved backwards via the learned vector field to obtain a latent noise sample $\boldsymbol{Z}_{\boldsymbol{O}_t}$, and $\|\boldsymbol{Z}_{\boldsymbol{O}_t}\|_2^2$ is used as uncertainty score.
- STAC [1] calculates the divergence between the temporally overlapping components of the action chunk distributions at two consecutive policy timesteps to detect temporally inconsistent behavior.
- RND-A is closely adapted from He et al. [22] and learns a confidence function using RND that measures the reliability of generated actions for successful task completion.

**Metrics.** We label successful rollouts as negative and failed ones as positive and define the true-positive-rate (TPR) and true-negative-rate (TNR) in the standard way. To account for imbalances in the numbers of successes and failures, we report the balanced accuracy $\mathrm{Acc} = \frac{1}{2}(\mathrm{TPR} + \mathrm{TNR})$. The normalized detection time $\mathrm{DT} = \min_t \{t|\, F(\boldsymbol{\tau}_{:t}) = 1\}/T \in [0,1]$ is calculated only for failed rollouts correctly flagged as `Fail`. We mark the DT if the TPR or TNR is below 0.4. In these cases, failures are either rarely detected or almost all rollouts are immediately flagged as `Fail`. Accuracy and DT are standard metrics in prior work [1, 75], but they are not ideal for evaluating failure prediction, which should be accurate *and* fast. Always waiting until the last rollout timestep before making a "prediction" on the outcome likely results in high accuracy, but is not suitable for predicting failures before they occur. Conversely, simply flagging each rollout as `Fail` in the first timestep would yield a perfect DT. Therefore, we propose timestep-wise accuracy (TWA) as a novel evaluation metric for failure prediction methods. TWA differs from accuracy in that a true positive is assigned a value $1 - \mathrm{DT}$ instead of 1, rewarding *earlier* correct failure predictions more strongly.

**Evaluation Protocol.** We use $M = 50$ successful rollouts for the three simulation environments and $M = 10$ for the two real-world tasks to train the learning-based failure predictors and calibrate the thresholds. Since there is generally no "best" quantile value for calibration (see Appendix C.4), we average all results over $1 - \delta \in \{0.9, 0.91, \ldots, 0.99\}$. We treat the window size and threshold type (CP band, CP constant or time-varying) as method-specific hyperparameters. Thus, we use the value $w_{A,O} \in \{1, \ldots, 50\}$ and threshold type that achieve the highest TWA across all environments, which we argue represents the best tradeoff between prediction accuracy, low DT, and robustness.

## 6 Results and Discussion

**Our proposed scores can distinguish failures from OOD.** Although they are often correlated, accurate failure prediction requires differentiating between OOD situations to which the policy can generalize and actual failures. To evaluate the effectiveness of our proposed observation- and action-based uncertainty scores in making this distinction, we divide rollouts into four categories, depending on their outcome and whether we have introduced OOD scenarios (see Section 5). We

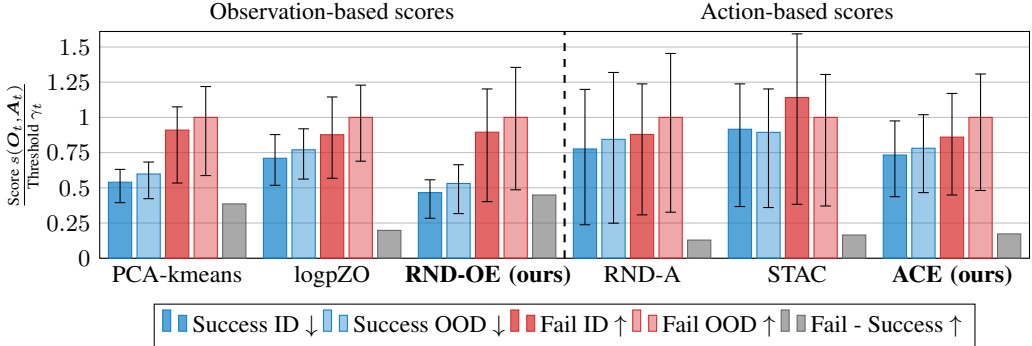

Figure 4: Our proposed scores (1) and (4) can distinguish failures from benign out-of-distribution (OOD) situations that the policy can generalize to. We group the rollouts into four categories along two axes: Success vs. Fail, and in-distribution (ID) vs. OOD. Robust failure prediction requires distinguishing Success OOD from Fail ID. We report the mean values across all tasks and five seeds, with the bars indicating the 25/75% quantiles.

Table 1: FIPER achieves superior failure prediction performance than the baselines. We highlight the best and underline the second-best value in each row. Detection times (DT) in brackets have corresponding TPR or TNR below 0.4, both of which indicate poor discrimination between successes and failures. We report the mean and standard deviation across five seeds.

| Task | Metric | PCA-kmeans [42] | logpZO [75] | RND-A [22] | STAC [1] | RND-OE (ours) | ACE (ours) | **FIPER (ours)** |
|---|---|---|---|---|---|---|---|---|
| SORTING | TWA ↑ | $0.49^{\pm0.00}$ | $\underline{0.54}^{\pm0.00}$ | $\mathbf{0.55}^{\pm0.01}$ | $0.46^{\pm0.00}$ | $0.52^{\pm0.01}$ | $\underline{0.54}^{\pm0.00}$ | $\underline{0.54}^{\pm0.00}$ |
| | Acc. ↑ | $0.56^{\pm0.00}$ | $\mathbf{0.67}^{\pm0.00}$ | $0.62^{\pm0.01}$ | $0.48^{\pm0.00}$ | $0.59^{\pm0.02}$ | $0.65^{\pm0.00}$ | $\underline{0.66}^{\pm0.00}$ |
| | DT ↓ | $(0.15^{\pm0.00})$ | $0.48^{\pm0.01}$ | $0.34^{\pm0.03}$ | $(0.46^{\pm0.00})$ | $(0.17^{\pm0.03})$ | $\mathbf{0.30}^{\pm0.00}$ | $\underline{0.32}^{\pm0.00}$ |
| STACKING | TWA ↑ | $\mathbf{0.66}^{\pm0.00}$ | $0.58^{\pm0.00}$ | $0.50^{\pm0.00}$ | $0.59^{\pm0.00}$ | $\underline{0.62}^{\pm0.04}$ | $\underline{0.62}^{\pm0.00}$ | $\underline{0.62}^{\pm0.00}$ |
| | Acc. ↑ | $\mathbf{0.75}^{\pm0.00}$ | $0.69^{\pm0.01}$ | $0.56^{\pm0.01}$ | $0.66^{\pm0.00}$ | $0.70^{\pm0.06}$ | $\underline{0.73}^{\pm0.00}$ | $\underline{0.73}^{\pm0.00}$ |
| | DT ↓ | $\underline{0.19}^{\pm0.00}$ | $0.49^{\pm0.00}$ | $(0.44^{\pm0.01})$ | $(0.38^{\pm0.00})$ | $\mathbf{0.16}^{\pm0.05}$ | $0.27^{\pm0.00}$ | $0.28^{\pm0.00}$ |
| PUSHT | TWA ↑ | $0.53^{\pm0.00}$ | $0.52^{\pm0.00}$ | $0.52^{\pm0.01}$ | $\mathbf{0.58}^{\pm0.00}$ | $0.54^{\pm0.01}$ | $\underline{0.56}^{\pm0.00}$ | $0.55^{\pm0.00}$ |
| | Acc. ↑ | $0.58^{\pm0.00}$ | $0.55^{\pm0.00}$ | $0.55^{\pm0.01}$ | $0.71^{\pm0.00}$ | $0.55^{\pm0.01}$ | $\underline{0.71}^{\pm0.00}$ | $\mathbf{0.71}^{\pm0.00}$ |
| | DT ↓ | $(0.11^{\pm0.00})$ | $(0.26^{\pm0.00})$ | $(0.20^{\pm0.02})$ | $0.52^{\pm0.00}$ | $(0.02^{\pm0.00})$ | $\mathbf{0.31}^{\pm0.00}$ | $\underline{0.32}^{\pm0.00}$ |
| PRETZEL | TWA ↑ | $0.64^{\pm0.00}$ | $0.58^{\pm0.03}$ | $0.51^{\pm0.05}$ | $0.51^{\pm0.00}$ | $0.55^{\pm0.02}$ | $\mathbf{0.75}^{\pm0.00}$ | $\underline{0.68}^{\pm0.03}$ |
| | Acc. ↑ | $0.65^{\pm0.00}$ | $0.65^{\pm0.04}$ | $0.53^{\pm0.05}$ | $0.67^{\pm0.00}$ | $0.72^{\pm0.02}$ | $\underline{0.82}^{\pm0.00}$ | $\mathbf{0.85}^{\pm0.00}$ |
| | DT ↓ | $(0.01^{\pm0.00})$ | $\underline{0.24}^{\pm0.04}$ | $(0.52^{\pm0.36})$ | $0.44^{\pm0.00}$ | $0.44^{\pm0.02}$ | $\mathbf{0.13}^{\pm0.00}$ | $0.33^{\pm0.07}$ |
| PUSHCHAIR | TWA ↑ | $0.50^{\pm0.00}$ | $\underline{0.78}^{\pm0.02}$ | $0.71^{\pm0.05}$ | $0.73^{\pm0.00}$ | $0.74^{\pm0.11}$ | $0.69^{\pm0.00}$ | $\mathbf{0.83}^{\pm0.02}$ |
| | Acc. ↑ | $0.50^{\pm0.00}$ | $\underline{0.92}^{\pm0.02}$ | $0.82^{\pm0.05}$ | $0.88^{\pm0.00}$ | $0.80^{\pm0.14}$ | $0.80^{\pm0.00}$ | $\mathbf{0.96}^{\pm0.02}$ |
| | DT ↓ | $(0.00^{\pm0.00})$ | $0.26^{\pm0.01}$ | $\underline{0.21}^{\pm0.02}$ | $0.30^{\pm0.00}$ | $\mathbf{0.11}^{\pm0.07}$ | $0.23^{\pm0.00}$ | $0.27^{\pm0.00}$ |
| Average | TWA ↑ | $0.57^{\pm0.00}$ | $0.60^{\pm0.01}$ | $0.56^{\pm0.02}$ | $0.57^{\pm0.00}$ | $0.59^{\pm0.04}$ | $\underline{0.63}^{\pm0.00}$ | $\mathbf{0.65}^{\pm0.01}$ |
| | Acc. ↑ | $0.61^{\pm0.00}$ | $0.69^{\pm0.01}$ | $0.62^{\pm0.03}$ | $0.68^{\pm0.00}$ | $0.67^{\pm0.05}$ | $\underline{0.74}^{\pm0.00}$ | $\mathbf{0.78}^{\pm0.00}$ |
| | DT ↓ | $(0.09^{\pm0.00})$ | $0.35^{\pm0.01}$ | $0.34^{\pm0.09}$ | $0.42^{\pm0.00}$ | $\mathbf{0.18}^{\pm0.03}$ | $\underline{0.25}^{\pm0.00}$ | $0.30^{\pm0.02}$ |

expect the uncertainty scores to increase in the following order: Success ID $\leq$ Success OOD $<$ Fail ID $\leq$ Fail OOD. In particular, the gap between Success OOD and Fail ID provides information about a score's ability to distinguish between OOD and failure, which affects the robustness of any failure predictor method using the score. We provide the average scores, divided by the respective CP band threshold, across all tasks in Figure 4. RND-OE and ACE show a clear separation between Success OOD and Fail ID. Compared to the other observation-based methods, PCA-kmeans and logpZO, our RND-OE score yields a larger difference between the mean values for Success OOD and Fail ID. The same holds for our ACE score compared to RND-A and STAC. However, the gap between Fail and Success is generally smaller for the action-based scores, indicating that failures are harder to detect from the policy outputs than from the inputs. In the following, we investigate whether the results regarding single-timestep uncertainty scores transfer to failure prediction performance.

**FIPER predicts failures more accurately and earlier.** We benchmark FIPER against four baselines and also include our individual observation- and action-based failure predictors (3) and (6). Table 1 summarizes the results of our evaluation. A more detailed version, including TPR and TNR, is

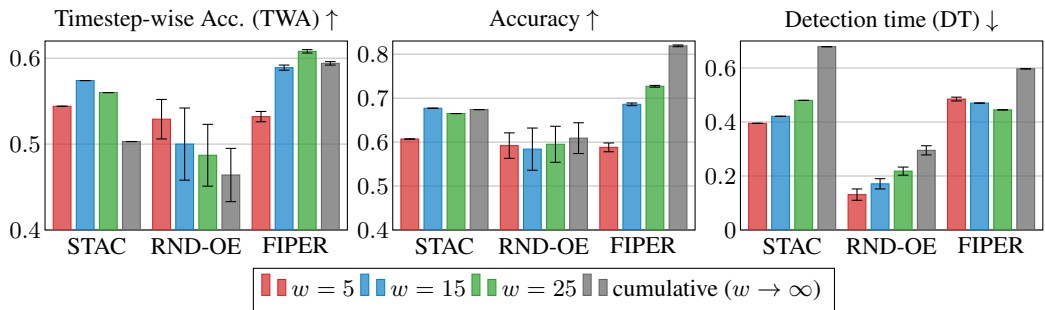

Figure 5: We compare our approach of aggregating uncertainty over a sliding window to accumulating scores over all past timesteps [1]. The latter method detects failure rollouts very late, mostly due to their greater length, whereas our approach can *predict* failures earlier. We use the same *CP constant* threshold definition for all methods, following STAC [1]. We normalize DT by the maximum episode length and average the results across five seeds, with the bars indicating the standard deviation.

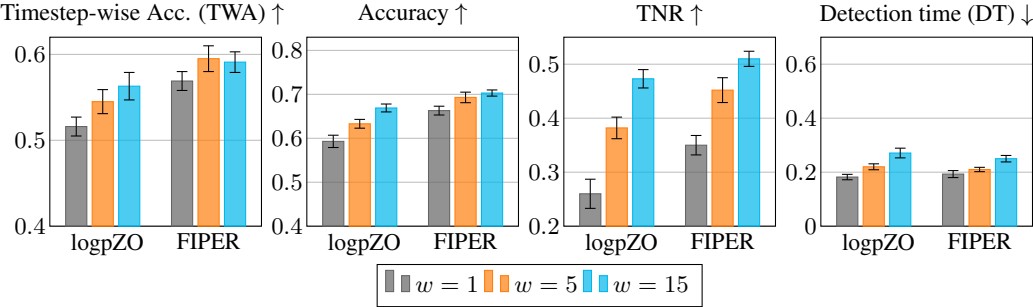

Figure 6: We compare our approach of aggregating uncertainty over a sliding window to making a decision based only on the current timestep, as proposed for logpZO [75]. Our method is much more robust to isolated OOD timesteps and greatly reduces false alarms (higher TNR) while only slightly increasing DT and improving overall accuracy. This comparison uses a *time-varying* threshold for all methods. We normalize DT by the maximum episode length and average the results across five seeds, with the bars indicating the standard deviation.

provided in Appendix C.2. FIPER obtains the highest TWA of 0.65 and accuracy of 0.78, and a lower DT of 0.30 than the baselines. Our two individual failure-predictors, RND-OE and ACE, can achieve even faster failure prediction when used alone, albeit with a lower TWA and accuracy. An overall TPR of 0.92 indicates that FIPER can predict different types of failures across diverse environments with high reliability. Compared to STAC, which also considers the distribution of generated actions, ACE predicts failures much more accurately, especially in the SORTING, STACKING, and PRETZEL environments that involve a high degree of action multimodality (see Figure 3 (d)). Among the observation-based methods, RND-OE achieves by far the lowest DT while performing better than PCA-kmeans and comparable to logpZO in terms of TWA and accuracy. PCA-kmeans is largely unable to distinguish OOD from failure, achieving a TNR of only 0.24, which shows that a larger difference between the average scores in Success and Fail rollouts (see Figure. 4) does not necessarily correlate with superior failure prediction performance. The baseline comparison demonstrates that FIPER represents the best combination of the desired properties of a failure predictor, i.e., high accuracy and early warnings.

**Aggregating uncertainty scores over multiple timesteps robustly predicts failures.** The two prior works most similar to ours either accumulate scores over all previous timesteps [1] or only use data from the current timestep [75]. We compare both approaches to our proposed method of aggregating scores over a sliding window. As the use of cumulative scores was proposed along with STAC [1], we use this baseline in the first comparison. The results shown in Figure 5 demonstrate that score accumulation can increase accuracy, but at the cost of much slower detection. In fact, we observe that cumulative uncertainty scores perform much better on tasks with a large difference in duration between successful and failed rollouts. Thus, we conclude that this approach recognizes

Table 2: Impact of the logical combination of the individual observation- and action-based failure predictors (3) and (6) of FIPER and the threshold calculation. The logical conjunction yields higher TWA and accuracy, showing that most failures are characterized by OOD observations *and* high entropy in generated actions. We report the mean and standard deviation across five seeds.

| Operator | Threshold | TWA ↑ | Acc. ↑ | DT ↓ | TPR ↑ | TNR ↑ |
|---|---|---|---|---|---|---|
| **AND** | CP band | $0.62^{\pm 0.00}$ | $\mathbf{0.78}^{\pm 0.01}$ | $0.47^{\pm 0.00}$ | $0.72^{\pm 0.01}$ | $0.84^{\pm 0.00}$ |
| | CP constant | $0.61^{\pm 0.00}$ | $0.73^{\pm 0.03}$ | $0.45^{\pm 0.00}$ | $0.59^{\pm 0.00}$ | $0.87^{\pm 0.00}$ |
| | time-varying | $\mathbf{0.65}^{\pm 0.01}$ | $\mathbf{0.78}^{\pm 0.00}$ | $0.30^{\pm 0.02}$ | $0.92^{\pm 0.00}$ | $0.65^{\pm 0.01}$ |
| OR | CP band | $0.54^{\pm 0.03}$ | $0.58^{\pm 0.03}$ | $(0.08^{\pm 0.02})$ | $0.98^{\pm 0.02}$ | $0.18^{\pm 0.08}$ |
| | CP constant | $0.59^{\pm 0.04}$ | $0.68^{\pm 0.05}$ | $\mathbf{0.20}^{\pm 0.03}$ | $0.93^{\pm 0.03}$ | $0.43^{\pm 0.12}$ |
| | time-varying | $0.51^{\pm 0.01}$ | $0.53^{\pm 0.01}$ | $(0.02^{\pm 0.01})$ | $1.00^{\pm 0.00}$ | $0.05^{\pm 0.02}$ |

failure rollouts primarily due to their greater length, which naturally leads to a higher cumulative score. In comparison, considering only the most recent timesteps enables us to actually *predict* failures and raise a warning early. This capability is especially important in safety-critical scenarios such as surgical assistance or collaborative assembly, even if it results in more false alarms. We also investigate using a single-timestep score (i.e., $w = 1$), as proposed for logpZO by FAIL-Detect [75]. As shown in Figure 6, this leads to a very low TNR, i.e., flagging most successful rollouts as `Fail`, while only slightly reducing DT. The overall failure prediction performance, as measured by TWA, improves when using a sliding window. Note that Figures 5 and 6 are not intended as a comparison of different failure prediction scores (e.g., STAC vs. RND-OE), as we fix the window size and threshold type. In summary, the results underline that aggregating uncertainty scores over the most recent timesteps is crucial for predicting failures at an early stage and distinguishing them from successes.

**Failures manifest in observations *and* actions.** FIPER predicts a failure if and only if both the observation- *and* action-based scores exceed their thresholds. We ablate the two design decisions of logical combination and threshold type, and report the results in Table 2. The logical conjunction (AND) yields higher TWA and accuracy, primarily due to its much higher TNR compared to OR. Intuitively, waiting for both scores to exceed their threshold increases the robustness to OOD success cases. Yet, we find that this more conservative approach can nonetheless predict 92% of all failures. This supports our claim that policy failure is associated with simultaneous OOD observations and high entropy in generated actions. As a variation of FIPER, the use of a logical disjunction of our two failure predictors provides a powerful alternative when the primary goal is to achieve very low DT and high TPR. This may be particularly desirable if failures could directly endanger people or cause damage to expensive items. Moreover, we observe that the threshold type has a significant impact on performance. The CP band and CP constant thresholds achieve a very high TNR in accordance with Proposition 1. The time-varying threshold yields the highest TWA for FIPER, while the inherently more sensitive OR ablation should be used with the more conservative CP constant threshold.

## 7 Conclusions and Limitations

The design of FIPER is motivated by the insight that actual task failures of generative diffusion- and flow-based policies are associated with successive OOD observations and high-uncertainty conditional action distributions. Our experiments across diverse environments and tasks demonstrate that our proposed observation- and action-based uncertainty scores can distinguish OOD from failure situations, and that FIPER can predict failures more accurately and earlier than existing methods. FIPER makes no assumptions about specific failure modes and requires no failure data, enabling its use in diverse human-centered environments where interpretable and safe robot behavior is critical.

While we use only a few successful rollouts for calibration, the need to collect them and train an RND-OE model that is separate from the policy is still a limitation of our approach. FIPER requires little runtime computation in our considered environments, but this could change with a very high-dimensional action space (e.g., for humanoid robots). We consider only single-task vision-based IL policies in this work. Adapting FIPER to recent large-scale VLAs [33, 6, 5], additional input modalities or reinforcement learning with generative policies represent interesting avenues for future work. Since our approach to failure prediction does not rely on access to the training data of the policy and operates directly in the observation embedding space, we expect it to work well for such potential extensions as well.

## Acknowledgements

Ralf Römer gratefully acknowledges the support of the research group ConVeY funded by the German Research Foundation under grant GRK 2428. This work has been supported by the Robotics Institute Germany, funded by BMFTR grant 16ME0997K.

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

# Appendix

## Table of Contents

# A Method Details

Below, we have summarized the most important mathematical symbols used in this work.

- $T$: Maximum episode length
- $k = 0, 1, \ldots, T$: Timestep
- $\boldsymbol{o}_k$: Robot observation
- $\boldsymbol{a}_k$: Robot action
- $T_\mathrm{h}$: Observation history length
- $H$: Action chunk length
- $h$: Action execution steps
- $t = 0, h, \ldots, T$: Policy timestep
- $\boldsymbol{O}_t$: Observation history
- $\boldsymbol{A}_t$: Action chunk
- $\mathbf{A}_t$: Batch of action chunks
- $B$: Action-chunk batch size
- $\pi(\boldsymbol{A}|\boldsymbol{O})$: Generative IL policy
- $\boldsymbol{\tau}_{:t}$: Trajectory of observation-action pairs up to policy timestep $t$
- $\boldsymbol{f_\theta}(\boldsymbol{O}_t)$: Trainable RND predictor network
- $\boldsymbol{g}(\boldsymbol{O}_t)$: Frozen RND target network
- $s_\mathrm{RND}(\boldsymbol{O}_t)$: RND observation embedding (RND-OE) score
- $\hat{E}(\cdot)$: Entropy of an action prediction step
- $s_\mathrm{ACE}(\mathbf{A}_t)$: Action-chunk entropy (ACE) score
- $w_O$: Window size for aggregating $s_\mathrm{RND}(\boldsymbol{O}_t)$
- $w_A$: Window size for aggregating $s_\mathrm{ACE}(\mathbf{A}_t)$
- $\eta_O(\boldsymbol{\tau}_{:t})$: Observation-based failure prediction score at policy timestep $t$
- $\eta_A(\boldsymbol{\tau}_{:t})$: Action-based failure prediction score at policy timestep $t$
- $\gamma_{O,t}$: Threshold for $\eta_O(\boldsymbol{\tau}_{:t})$
- $\gamma_{A,t}$: Threshold for $\eta_A(\boldsymbol{\tau}_{:t})$
- $\mathcal{D}_\mathrm{c}$: Calibration dataset
- $M$: Number of calibration rollouts
- $q_\pi$: Distribution of ID successful policy rollouts
- $\delta$: Desired FPR

## A.1 Action-Chunk Entropy (ACE)

To estimate uncertainty in generated actions, we sample a batch of $B$ action chunks

$$\mathbf{A}_t = \big(\boldsymbol{A}_t^{(1)}, \ldots, \boldsymbol{A}_t^{(B)}\big), \quad \boldsymbol{A}_t^{(j)} = \big(\boldsymbol{a}_{t|t}^{(j)}, \ldots, \boldsymbol{a}_{t+H-1|t}^{(j)}\big) \sim \pi(\cdot \mid \boldsymbol{O}_t), \tag{9}$$

at policy timestep $t$ and compute the action-chunk entropy (ACE) score

$$s_\mathrm{ACE}(\mathbf{A}_t) = \sum_{i=0}^{H-1} \hat{E}\big(\boldsymbol{a}_{t+i|t}^{(1)}, \ldots, \boldsymbol{a}_{t+i|t}^{(B)}\big), \tag{10}$$

where $\hat{E}(\cdot)$ measures the entropy of an action prediction step $t + i$ based on the actions $\boldsymbol{a}_{t+i|t}^{(j)} \in \mathcal{A} \subseteq \mathbb{R}^D$, $j = 1, \ldots, B$, sampled for that timestep. For the following explanation of the calculation of $\hat{E}(\cdot)$, we use the notation $\boldsymbol{a}_{t+i|t}^{(j)} = \boldsymbol{a}_{t+i}^{(j)} = \big[a_{t+i,1}^{(j)}, \ldots, a_{t+i,D}^{(j)}\big]^\mathsf{T}$

for brevity. Let $\{\boldsymbol{A}_t, \ldots, \boldsymbol{A}_{N_c t}\}$ be the set of action chunks in the calibration dataset $\mathcal{D}_{\mathrm{c}}$, where $\boldsymbol{A}_{nt} = (\boldsymbol{a}_{nt}, \ldots, \boldsymbol{a}_{nt+H-1})$, $n = 1, \ldots, N_{\mathrm{c}}$. Offline, we calculate the maximum range of actions as

$$R_d = \max_{\substack{n \in \{1, \ldots, N_c\} \\ i \in \{0, \ldots, H-1\}}} a_{nt+i,d} - \min_{\substack{n \in \{1, \ldots, N_c\} \\ i \in \{0, \ldots, H-1\}}} a_{nt+i,d} \tag{11}$$

for each dimension $d = 1, \ldots, D$ and define the fixed cell size as $\alpha R_d$, where $\alpha \in (0,1)$ is called cell size factor. For a new action batch (9), each action dimension $d$ is partitioned into

$$N_d^i = \left\lceil \frac{\max_{j \in \{1, \ldots, B\}} a_{t+i,d}^{(j)} - \min_{j \in \{1, \ldots, B\}} a_{t+i,d}^{(j)}}{\alpha R_d} \right\rceil \tag{12}$$

bins for each step $i = 0, \ldots, H-1$ in the prediction horizon. This induces $N^i = \prod_{d=1}^{D} N_d^i$ bin cells, which we denote by $\mathcal{C}_c^i \subset \mathbb{R}^D$, $c = 1, \ldots, N^i$. We then build timestep-wise histograms for the $B$ actions sampled for each timestep $t + i$ over these cells, yielding the probabilities $p_c^i = \frac{n_c^i}{B}$, where $n_c^i = \left| j : \boldsymbol{a}_{t+i}^{(j)} \in \mathcal{C}_c^i \right|$ is the number of actions for timestep $t + i$ that are contained in cell $\mathcal{C}_c^i$. Finally, the action entropy for each timestep in (10) is given by

$$\hat{E}\big(\boldsymbol{a}_{t+i|t}^{(1)}, \ldots, \boldsymbol{a}_{t+i|t}^{(B)}\big) = -\sum_{c=1}^{N^i} p_c^i \log_2 p_c^i. \tag{13}$$

Our approach of treating timesteps separately leverages the fact that actions sampled for the same timestep tend to lie closer together, significantly reducing the total number of bins. Thus, compared to computing a single histogram over all timesteps, we require a much smaller action batch size to obtain a sufficiently accurate estimate of action entropy.

We calculate the ACE score in the Cartesian space using predicted end-effector positions. This is possible for any manipulation task (and even for locomotion or navigation) by using forward kinematics, even if the policy outputs joint angles or velocities. We adopt this approach not only for computational reasons but also to obtain interpretable uncertainty scores in the same 3D space where the task is performed. High uncertainty in Cartesian space means that the policy is unsure about the correct path to take (e.g., where to grasp an object). In contrast, entropy in, for example, joint velocities, does not necessarily correspond to task-relevant uncertainty. We also experimented with computing the ACE score using other action representations, such as full 6D poses, but did not observe improvements in performance.

## A.2 Random Network Distillation with Observation Embeddings (RND-OE)

An RND model [9] detects OOD situations by training a predictor network $\boldsymbol{f_\theta}(\cdot)$ to match the outputs of a frozen target network $\boldsymbol{g}(\cdot)$ on ID data. Both networks receive the same inputs, and after training, the outputs of the predictor and target networks typically differ more significantly for inputs that deviate from the training distribution than for those that align with it. Our RND-OE model utilizes the observation embeddings from the encoder module of a generative policy as input and is trained on a calibration dataset that exclusively consists of a few ID successful policy rollouts. Thereby, we employ a multi-layer fully connected neural network architecture for both the target and predictor networks of the RND-OE model. Both networks take observation embeddings as input and output an $m$-dimensional feature vector. The target network $\boldsymbol{g}$ comprises four fully connected layers with dimensions $\dim(\boldsymbol{O}^{\mathrm{e}}) \to 1024 \to 2048 \to 4096 \to m$, each followed by a LeakyReLU activation function except for the last one. Its weights are randomly initialized and frozen before training. The predictor network $\boldsymbol{f_\theta}$ mirrors the target network's first three layers but includes two additional fully connected layers with ReLU activation and decreasing dimensions, i.e., $\dim(\boldsymbol{O}^{\mathrm{e}}) \to 1024 \to 2048 \to 4096 \to 2048 \to 1024 \to m$. The RND score is calculated as the pairwise distance between the outputs of the predictor and the target network. We use the same network architecture across all environments, and only the input dimension $\dim(\boldsymbol{O}^{\mathrm{e}})$ varies between the environments. The training parameters for the RND-OE models are given in Table 3.

## A.3 Conformal Prediction

For clarity, we restate the conformal prediction (CP) problem we consider for failure prediction in the following. We assume the availability of a calibration dataset $\mathcal{D}_{\mathrm{c}} = \{\boldsymbol{\tau}^i\}_{i=1}^{M} \sim q_\pi$ of $M$ independent

Table 3: Training parameters of the RND-OE model.

| Parameter | Value |
|---|---|
| Batch size | 256 |
| Epochs | 250 |
| Learning rate | $1 \times 10^{-4}$ |
| Learning rate scheduler | cosine |
| Optimizer | AdamW |
| Optimizer weight decay | $1 \times 10^{-5}$ |
| Optimizer epsilon | $1 \times 10^{-8}$ |
| Train/validation split | 0.9/0.1 |

and identically distributed (i.i.d) successful policy rollouts and define $\mathcal{I} = \{1, \dots, M\}$. For a new successful rollout $\boldsymbol{\tau} = (\boldsymbol{O}_0, \boldsymbol{A}_0, \boldsymbol{O}_h, \boldsymbol{A}_h, \dots, \boldsymbol{O}_{T'}, \boldsymbol{A}_{T'}) \sim q_\pi$ from the same distribution, we aim for an upper bound $\delta$ on the probability that this rollout is incorrectly flagged as `Fail`, i.e.,

$$\mathrm{P}\big(\exists t \in \{0, h, \dots, T'\} \text{ s.t. } \eta(\boldsymbol{\tau}_{:t}) > \gamma_t\big) \leq \delta, \tag{14}$$

where $\eta(\boldsymbol{\tau}_{:t})$ and $\gamma_t$ are the score function and corresponding threshold of the failure predictor $F(\boldsymbol{\tau}_{:t}) = \mathbb{1}(\eta(\boldsymbol{\tau}_{:t}) > \gamma_t)$ (see Sections 4.1 and 4.2). To achieve this, the threshold $\gamma_t$ needs to be defined using a suitable nonconformity measure. A simple (but conservative) approach is to consider the maximum failure prediction score over an entire calibration rollout

$$\bar{\eta}(\boldsymbol{\tau}^i) = \max_t \ \eta(\boldsymbol{\tau}^i_{:t}), \qquad i \in \mathcal{I}. \tag{15}$$

**Theorem 1** (Adapted from [3], Theorem D.1). *Let $\boldsymbol{\tau}^1, \dots, \boldsymbol{\tau}^M$ and $\boldsymbol{\tau}$ be i.i.d.. Then, defining*

$$\gamma = \inf \left\{ \beta : \frac{|\{i \in \mathcal{I} : \bar{\eta}(\boldsymbol{\tau}^i) \leq \beta\}|}{M} \geq \frac{\lceil (M+1)(1-\delta) \rceil}{M} \right\} \tag{16}$$

*as the empirical $\frac{\lceil (M+1)(1-\delta) \rceil}{M}$ quantile of $\{\bar{\eta}(\boldsymbol{\tau}^i) : i \in \mathcal{I}\}$ ensures that*

$$\mathrm{P}(\bar{\eta}(\boldsymbol{\tau}) \leq \gamma) \geq 1 - \delta. \tag{17}$$

**Proposition 2** (Bounded FPR with a CP constant threshold). *Suppose the calibration rollouts $\boldsymbol{\tau}^1, \dots, \boldsymbol{\tau}^M$ and the test rollout $\boldsymbol{\tau}$ are i.i.d., and let $\gamma$ be defined according to (16). Then, setting $\gamma_t = \gamma$ ensures that (14) holds.*

*Proof.* The definition of the nonconformity score (15) implies that

$$\bar{\eta}(\boldsymbol{\tau}) = \max_{\tilde{t}} \ \eta(\boldsymbol{O}_0, \boldsymbol{A}_0, \dots, \boldsymbol{O}_{\tilde{t}}, \boldsymbol{A}_{\tilde{t}}) \geq \eta(\boldsymbol{O}_0, \boldsymbol{A}_0, \dots, \boldsymbol{O}_t, \boldsymbol{A}_t) = \eta(\boldsymbol{\tau}_{:t}), \quad \forall t \in \{0, h, \dots, T'\}. \tag{18}$$

Therefore,

$$\mathrm{P}\big(\exists t \in \{0, h, \dots, T'\} \text{ s.t. } \eta(\boldsymbol{\tau}_{:t}) > \gamma_t\big) \leq \mathrm{P}\big(\exists t \in \{0, h, \dots, T'\} \text{ s.t. } \bar{\eta}(\boldsymbol{\tau}) > \gamma_t\big) \tag{19}$$

$$= \mathrm{P}\big(\bar{\eta}(\boldsymbol{\tau}) > \gamma\big) \tag{20}$$

$$= 1 - \mathrm{P}\big(\bar{\eta}(\boldsymbol{\tau}) \leq \gamma\big) \tag{21}$$

$$\leq \delta, \tag{22}$$

where we have used $\gamma_t = \gamma$ in (20) and Theorem 1 in (22). $\qquad\square$

We can obtain a tighter set of thresholds by defining a CP band, as described by Diquigiovanni et al. [15]. For clarity, we define $\eta_i(t) = \eta(\boldsymbol{\tau}^i_{:t})$ and $\eta(t) = \eta(\boldsymbol{\tau}_{:t})$. We split the set $\mathcal{I}$ into two disjoint parts, $\mathcal{I}_1$ and $\mathcal{I}_2$, of sizes $M_1$ and $M_2 = M - M_1$, respectively. The first set is used to calculate the sample functional mean as

$$\mu(t) = \frac{1}{M_1} \sum_{i=1}^M \mathbb{1}(i \in \mathcal{I}_1) \eta_i(t), \qquad t = 0, 1, \dots, T. \tag{23}$$

The maximum deviation from the mean for a calibration rollout $i \in \mathcal{I}_2$, scaled by a modulation function $s(t)$ such as, for example, $s(t) = 1/T$, is given by

$$R_i^s = \max_t \left| \frac{\eta_i(t) - \mu(t)}{s(t)} \right|. \tag{24}$$

**Theorem 2** (Adapted from [15], Appendix A.3). *Let $\boldsymbol{\tau}^1, \ldots, \boldsymbol{\tau}^M$ and $\boldsymbol{\tau}$ be i.i.d., and let $\mu(t)$ be the sample functional mean defined according to (23). Then, defining $k^s$ as the empirical $1 - \delta$ quantile of $\{R_i^s : i \in \mathcal{I}_2\}$ guarantees that*

$$\mathrm{P}\big(\eta(t) \in [\mu(t) - k^s s(t), \mu(t) + k^s s(t)], \ \forall t\big) \geq 1 - \delta. \tag{25}$$

For more information on the construction of alternative modulation functions $s(t)$, refer to Diquigiovanni et al. [15].

**Proposition 3** (Bounded FPR with a one-sided CP band). *Suppose the calibration rollouts $\boldsymbol{\tau}^1, \ldots, \boldsymbol{\tau}^M$ and the test rollout $\boldsymbol{\tau}$ are i.i.d., and let $\mu(t)$ and $k^s$ be defined according to (23) and Theorem 2, respectively. Then, setting $\gamma_t = \mu(t) + k^s s(t)$ ensures that (14) holds.*

*Proof.* Theorem 2 implies that

$$1 - \delta \leq \mathrm{P}(\eta(t) \in [\mu(t) - k^s s(t), \mu(t) + k^s s(t)], \ \forall t) \tag{26}$$
$$\leq \mathrm{P}(\eta(t) \in (-\infty, \mu(t) + k^s s(t)], \ \forall t) \tag{27}$$
$$= \mathrm{P}(\eta(t) \leq \mu(t) + k^s s(t), \ \forall t) \tag{28}$$

Moreover, we can rewrite the probability term in (14) as

$$\mathrm{P}\big(\exists t \in \{0, h, \ldots, T'\} \text{ s.t. } \eta(\boldsymbol{\tau}_{:t}) > \gamma_t\big) = 1 - \mathrm{P}\big(\eta(\boldsymbol{\tau}_{:t}) \leq \gamma_t, \ \forall t\big) \tag{29}$$

Substituting $\eta(\boldsymbol{\tau}_{:t}) = \eta(t)$ and $\gamma_t = \mu(t) + k^s s(t)$ and plugging (28) into (29) directly yields (14), which concludes the proof. $\square$

*Remark* 2. In addition to the one-sided CP band described above, we have also tested a simpler time-varying threshold $\gamma_t = \mu(t) + \chi(1 - \delta)\sigma(t)$, where $\chi(\cdot)$ is the inverse cumulative distribution function of the univariate Gaussian, and $\sigma(t)$ is the standard deviation of $\{\eta_i(t) : i \in \mathcal{I}_2\}$ with respect to $\mu(t)$. This threshold definition treats the rollout timesteps independently and avoids taking the maximum over $t$, as in (15) and (24), resulting in a more sensitive failure predictor that is more likely to raise warnings, albeit at the cost of an increase in false alarms. A performance comparison of all three threshold definitions is provided in Figure 12.

## A.4   Logical Combination

FIPER combines the outputs of the observation-based failure predictor (3) and the action-based failure predictor (6) with a logical conjunction (AND). Consequently, a rollout is only flagged as `Fail` if both predictors exceed their respective thresholds, ensuring high robustness against benign OOD situations that the policy can handle. If both thresholds $\gamma_{O,t}$ and $\gamma_{A,t}$ are separately calibrated according to Proposition 2 or Proposition 3, FIPER satisfies the same upper bound on the FPR as the individual failure predictors.

**Proposition 4** (Proposition 1 extended). *Set $\delta \in (0, 1)$, and define the thresholds $\gamma_{O,t}$ and $\gamma_{A,t}$ for the failure prediction scores $\eta_O(\boldsymbol{\tau}_{:t})$ and $\eta_A(\boldsymbol{\tau}_{:t})$ using conformal prediction according to Propositions 2 or 3. Then, the probability that the failure predictor (7) of FIPER incorrectly flags a new successful ID rollout $\boldsymbol{\tau} \sim q_\pi$ of length $T' \leq T$ as `Fail` at any policy timestep $t \leq T'$ satisfies the upper bound*

$$\mathrm{P}\big(\exists t \in \{0, h, \ldots, T'\} \text{ s.t. } F(\boldsymbol{\tau}_{:t}) = 1\big) \leq \delta. \tag{30}$$

*Proof.* For each timestep $t \in \{0, h, \ldots, T'\}$, we have

$$\mathrm{P}\big(F(\boldsymbol{\tau}_{:t}) = 1\big) = \mathrm{P}\big(F_O(\boldsymbol{\tau}_{:t}) = 1 \wedge F_A(\boldsymbol{\tau}_{:t}) = 1\big) \tag{31}$$
$$= \mathrm{P}\big(F_A(\boldsymbol{\tau}_{:t}) = 1\big)\mathrm{P}\big(F_O(\boldsymbol{\tau}_{:t}) = 1 \mid F_A(\boldsymbol{\tau}_{:t}) = 1\big) \tag{32}$$
$$\leq \mathrm{P}\big(F_A(\boldsymbol{\tau}_{:t}) = 1\big). \tag{33}$$

The result directly follows from the fact that $\mathrm{P}\big(\exists t \in \{0, h, \ldots, T'\} \text{ s.t. } F_A(\boldsymbol{\tau}_{:t}) = 1\big) \leq \delta$ by definition of $\gamma_{A,t}$. $\square$

As a natural alternative to the logical conjunction, we have also investigated a logical disjunction (OR) of the two outputs. This variant can predict *all* failures, but it often triggers false alarms, as shown in Table 2. Since FIPER also achieves a high TPR of 0.92 with the logical AND, combined with a much lower rate of false alarms and a higher TWA, we argue that the logical conjunction of the observation- and action-based failure predictor is the better choice for accurate and robust failure prediction. As an alternative to combining the detector outputs, we have also investigated a weighted combination of the normalized scores (1) and (4). However, this approach would add an additional (task-dependent) hyperparameter and did not show superior performance in our experiments.

## B  Additional Experimental Details

### B.1  Environment Details

Table 4: Details about our environments and datasets.

| Environment | SORTING | STACKING | PUSHT | PRETZEL | PUSHCHAIR |
|---|---|---|---|---|---|
| Type | sim | sim | sim | real-world | real-world |
| $\dim(\boldsymbol{o})$ | [2, 3, 96, 96] | [2, 3, 96, 96] | [3, 512, 512] | [3, 240, 320] | [3, 270, 480] |
| $\dim(\boldsymbol{O}^{\text{e}})$ | 128 | 128 | 64 | 512 | 96 |
| $\dim(\boldsymbol{a})$ | 2 | 8 | 2 | 5 | 3 |
| $\dim(\boldsymbol{s})$ | 3 | 8 | 2 | 5 | 13 |
| # Calibration rollouts | 50 | 50 | 50 | 10 | 10 |
| # Test rollouts | 400 | 800 | 300 | 20 | 20 |
| # Test rollouts (ID) | 100 | 200 | 150 | 0 | 0 |
| # Test rollouts (OOD) | 300 | 600 | 150 | 20 | 20 |
| Success rate (ID) | 0.89 | 0.60 | 0.40 | - | - |
| Success rate (OOD) | 0.57 | 0.36 | 0.29 | 0.5 | 0.67 |
| Avg. episode length | 47 | 79 | 31 | 55 | 8 |
| Max. episode length | 75 | 120 | 38 | 70 | 22 |

This section provides an overview of the environments we use to evaluate our failure prediction framework. We summarize the details about the environments and datasets in Table 4. For PUSHT and PUSHCHAIR, we use the publicly available rollout datasets from Agia et al. [1], which are released under an MIT License.

### B.1.1  Simulation Environments

The three simulation tasks, SORTING, STACKING, and PUSHT, are visualized in Figure 7.

SORTING:   In this task, a Franka robot must push two blocks into their corresponding color-matching target boxes on a tabletop. The visual observations are obtained from a wrist camera and a third-person camera. To induce more policy failures in our test data, we vary the dimensions of the blocks and the positions of the target boxes. A rollout is considered successful if the policy manages to place both blocks in their respective boxes within the maximum episode length. If one of the blocks falls from the table but does not land inside its corresponding box, it is impossible for the policy to correct this mistake; therefore, we end the rollout prematurely.

STACKING:   A Franka robot is trained to stack three colored blocks in a target area, using joint velocities and gripper commands as actions. Similar to SORTING, two RGB images are used as visual inputs. The overall complexity of this task leads to a very low success rate of the complete task (referred to as STACKING-3 in D3IL [29]), especially when introducing OOD scenarios. Therefore, we only consider the simpler STACKING-1 task for testing, which is successfully completed if one of the three blocks has been placed into the target area (see Figure 7). This allows us to include OOD situations by varying the dimensions of the blocks and the location of the target area, while still achieving a sufficiently high success rate to evaluate failure prediction. This task exhibits strong action multimodality, and failures often occur when the robot is unable to grasp a block.

PUSHT:   A planar T-shaped object needs to be pushed into a target configuration. We use the rollout data from Agia et al. [1], which includes variations of the scale and dimensions of the T-object. A rollout is considered successful if the policy manages to push the "T" into the goal area with at least

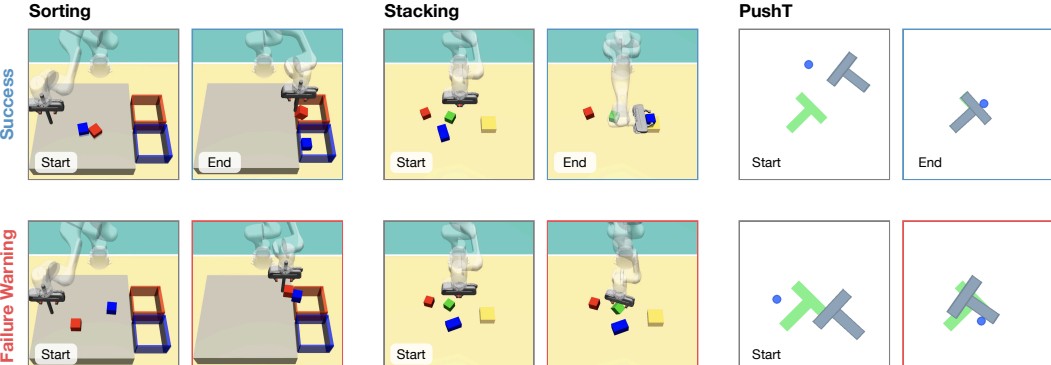

Figure 7: Overview of the simulation tasks with exemplary successful and failed rollouts.

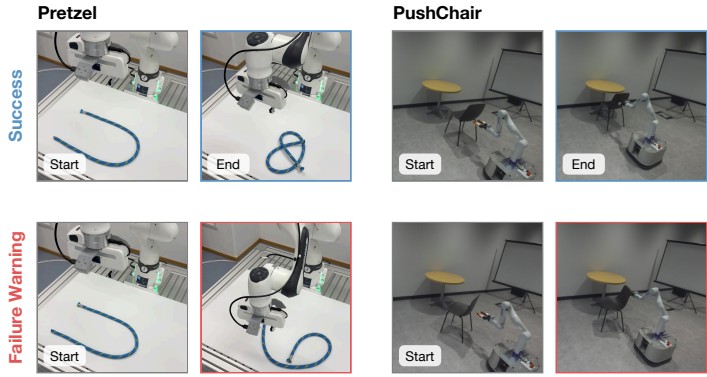

Figure 8: Overview of the real-world tasks with exemplary successful and failed rollouts.

90% overlap. Since the policy has learned to approach the "T" from different directions, this task exhibits strong action multimodality. Failures for this task can occur when the policy is unable to properly move the "T" due to its changed size, or if it gets stuck in a suboptimal pose, where the "T" somewhat overlaps with the target area but in an incorrect orientation (see failure case in Figure 7).

### B.1.2 Real-World Tasks

Figure 8 depicts the two real-world tasks considered in this work, namely Pretzel and PUSHCHAIR.

PRETZEL:    In this task, a Franka robot needs to form a rope into the shape of a pretzel. A successfully formed pretzel comprises three intersections of the rope that divide the area into three distinct parts when looking from above, as shown in the successful rollout in Figure 8. We count a rollout as successful if the robot manages to form a correct pretzel within 70 policy inference timesteps, otherwise, it is considered a failure. The maximum episode length is thereby determined by the longest successful rollout in our test set. To invoke more policy failures in our test data, we vary the rope's initial position and apply rotations about its axis. Because the rope's material is anisotropic, axial rotations can change its bending behavior. In that case, once the robot places the rope onto the table, the rope does not remain in this configuration but moves back toward its original pose (see failure warning in Figure 8). Unlike variations of the initial conditions, this OOD behavior can not be detected from the initial observation images, but only becomes apparent after multiple timesteps when one end of the rope has been placed. In our experiments, vertical and horizontal position perturbations lead to 6 and 2 failures, respectively, and axial rotations induce an additional 2 failures.

PUSHCHAIR:    A single-arm mobile manipulator needs to push a chair to a target position underneath a table. We use the data from Agia et al. [1], which includes variations of the initial chair pose. Failures occur when the chair rotates substantially about the vertical axis, which can happen, for example, if the robot does not push against the center of the backrest (see Figure 8).

Table 5: Implementation details for our generative IL policies based on flow matching (FM) [40] and denoising diffusion probabilistic models (DDPM) [25]. The policy hyperparameters for PushT and PushChair are taken from Agia et al. [1].

| Task | Sorting | Stacking | PushT | Pretzel | PushChair |
|---|---|---|---|---|---|
| Generative modeling technique | FM | FM | DDPM | DDPM | DDPM |
| Backbone | ACT [78] | ACT [78] | U-Net [11] | U-Net [11] | U-Net [11] |
| Observation history length $T_{\mathrm{h}}$ | 1 | 5 | 1 | 1 | 1 |
| Action chunk length $H$ | 8 | 8 | 16 | 16 | 16 |
| Action execution steps $h$ | 4 | 4 | 8 | 8 | 4 |
| Action batch size $B$ | 32 | 32 | 256 | 30 | 256 |
| Integration steps | 16 | 16 | 100 | 100 | 100 |
| Training epochs | 20 | 20 | 2000 | 100 | 2000 |
| Training batch size | 256 | 256 | 256 | 64 | 256 |
| Optimizer | AdamW | AdamW | AdamW | AdamW | AdamW |
| Learning rate | $5 \times 10^{-4}$ | $5 \times 10^{-4}$ | $3 \times 10^{-5}$ | $1 \times 10^{-4}$ | $3 \times 10^{-5}$ |

## B.2 Discussion on other Tasks

While we focus on manipulation in our experiments, we believe that FIPER is also directly applicable to other tasks, such as navigation and locomotion, provided that two conditions are met:

- There is a clear definition of "task failure". This might not be as obvious for some locomotion tasks, such as tracking a desired reference base velocity, which do not have the same episodic character as typical manipulation and navigation tasks. Intuitively, task failure is well-defined if there is a maximum episode length and a clear objective that should be achieved within that time frame. For locomotion, this objective could be defined, for example, as staying within a certain tolerance around the reference speed.

- The actions can be represented in or transformed to Cartesian space. This is not a strict requirement, but we think it is highly beneficial for the reasons mentioned in Appendix A.1. It applies to navigation, where the entropy can be computed for the base positions predicted from the action chunks. For locomotion, one possible approach would be to compute the action entropy for the positions of each foot individually and then combine these scores by averaging or taking the maximum.

## B.3 Policy Details

Given a demonstration dataset $\mathcal{D} = \{(\boldsymbol{A}_0, \boldsymbol{O}_0), (\boldsymbol{A}_1, \boldsymbol{O}_1), \dots\}$ of observation-action chunk pairs sampled from an unknown expert policy $q(\boldsymbol{A}|\boldsymbol{O})$, we aim to learn an IL policy $\pi(\boldsymbol{A}|\boldsymbol{O})$ that approximates the conditional action expert distribution as much as possible, i.e., $\pi(\boldsymbol{A}|\boldsymbol{O}) \approx q(\boldsymbol{A}|\boldsymbol{O})$. We consider IL policies based on the two most popular generative modeling techniques, diffusion and flow matching, and denote their learnable parameters by $\psi$. Details on the implementation of our policies are provided in Table 5. For SORTING and STACKING, we combine flow matching with a transformer backbone. For PUSHT, PRETZEL, and PUSHCHAIR, we combine diffusion with a CNN backbone. We have made this choice independently of the tasks' characteristics to evaluate our method's performance for different policy formulations and architectures.

### B.3.1 Diffusion Policy

Denoising diffusion probabilistic models (DDPM) [25] introduce latent variables $\boldsymbol{A}^0, \dots, \boldsymbol{A}^R$, where $\boldsymbol{A}^0 = \boldsymbol{A}$, and construct a forward diffusion Markov process

$$q\left(\boldsymbol{A}^r|\boldsymbol{A}^{r-1}, \boldsymbol{O}, r\right) = \mathcal{N}\left(\sqrt{1 - \beta_r}\boldsymbol{A}^{r-1}, \beta_r \boldsymbol{I}\right), \tag{34}$$

where $r = 1, \dots, R$ is the diffusion time step, and $\beta_{1:R}$ is a pre-defined noise schedule, such as the cosine noise schedule [48]. The noise schedule is chosen such that $q\left(\boldsymbol{A}^R\right) \approx \mathcal{N}(\boldsymbol{O}, \boldsymbol{I})$. The objective is to reverse the forward process by a learnable denoising (backward) process

$$\pi\left(\boldsymbol{A}^{r-1}|\boldsymbol{A}^r, \boldsymbol{O}, r\right) = \mathcal{N}\left(\boldsymbol{\mu}_{\boldsymbol{\psi}}\left(\boldsymbol{A}^r, \boldsymbol{O}, r\right), \sigma_r^2 \boldsymbol{I}\right), \tag{35}$$

where $\sigma_r^2 = \beta_r \frac{1-\alpha_{r-1}}{1-\alpha_r}$ with $\alpha_r = \prod_{i=1}^r (1 - \beta_i)$. This can be achieved by learning to predict the Gaussian noise added to $\boldsymbol{A}^0$ via the surrogate training loss

$$\mathcal{L}_{\mathrm{DP}}(\boldsymbol{\psi}) = \mathbb{E}_{r\sim\mathcal{U}(\{1,\ldots,R\}),(\boldsymbol{A}^0,\boldsymbol{O})\sim\mathcal{D},\boldsymbol{\epsilon}\sim\mathcal{N}(\mathbf{0},\boldsymbol{I})} \left[\left\|\boldsymbol{\epsilon} - \boldsymbol{\epsilon}_{\boldsymbol{\psi}}\left(\sqrt{\alpha_r}\boldsymbol{A}^0 + \sqrt{1-\alpha_r}\boldsymbol{\epsilon}, \boldsymbol{O}, r\right)\right\|_2\right]. \quad (36)$$

After training, a new action sequence conditioned on an observation $\boldsymbol{O}_t$ can be generated by first sampling $\boldsymbol{A}^R \sim \mathcal{N}(\mathbf{0}, \boldsymbol{I})$ and then iterating over the learned denoising process (35) with $\boldsymbol{\mu}_{\boldsymbol{\psi}}\left(\boldsymbol{A}^r, \boldsymbol{O}_t, r\right) = \frac{1}{\sqrt{1-\beta_r}}\left(\boldsymbol{A}^r - \frac{\beta_r}{\sqrt{1-\alpha_r}}\boldsymbol{\epsilon}_{\boldsymbol{\psi}}\left(\boldsymbol{A}^r, \boldsymbol{O}, r\right)\right)$ for $r = R, \ldots, 1$, resulting in $\boldsymbol{A}^0 \sim \pi(\cdot|\boldsymbol{O}_t)$.

### B.3.2 Flow Matching Policy

Denoising diffusion can be viewed as solving a stochastic differential equation (SDE) [66]. In contrast, flow matching [40] aims to learn the straight velocity field of an ordinary differential equation (ODE) that transports samples from a simple base distribution—typically a standard Gaussian— to the unknown target distribution. To be more consistent with Appendix B.3.1, we slightly deviate from the notation commonly used in the flow matching literature and denote samples from the source and target distribution by $\boldsymbol{A}^1 \sim q_1 = \mathcal{N}(\mathbf{0}, \boldsymbol{I})$ and $\boldsymbol{A}^0 \sim q_0 = q(\boldsymbol{A}|\boldsymbol{O})$, respectively, instead of the other way around. A flow $\frac{\mathrm{d}}{\mathrm{d}s}\boldsymbol{\phi}_s\left(\boldsymbol{A}\right) = \boldsymbol{u}_s\left(\boldsymbol{\phi}_s\left(\boldsymbol{A}\right)\right)$ determined by a velocity field $\boldsymbol{u}_s$ induces a probability path $q_s$ from $q_1$ to $q_0$ if $\boldsymbol{A}^s = \boldsymbol{\phi}_s\left(\boldsymbol{A}^1\right) \sim q_s$. Instead of directly approximating the unknown velocity field $\boldsymbol{u}_s$, flow matching is trained by conditioning on a target sample $\boldsymbol{A}^0 \sim q_0$ via the conditional flow matching loss

$$\mathcal{L}_{\mathrm{FM}}(\boldsymbol{\psi}) = \mathbb{E}_{s\sim\mathcal{U}([0,1]),(\boldsymbol{A}^0,\boldsymbol{O})\sim\mathcal{D},\boldsymbol{A}^s\sim q_s(\cdot|\boldsymbol{A}^0)} \left[\left\|\boldsymbol{v}_{\boldsymbol{\psi}}\left(\boldsymbol{A}^s, \boldsymbol{O}, s\right) - \boldsymbol{u}_s\left(\boldsymbol{A}^s|\boldsymbol{A}^0\right)\right\|_2\right]. \quad (37)$$

There are different ways to define the conditional velocity field $\boldsymbol{u}_s\left(\boldsymbol{A}^s|\boldsymbol{A}^0\right)$ [41]. We use a simple linear interpolation $\boldsymbol{A}^s = (1-s)\boldsymbol{A}^0 + s\boldsymbol{A}^1$ with $\boldsymbol{u}_s\left(\boldsymbol{A}^s|\boldsymbol{A}^0\right) = \frac{\boldsymbol{A}^0-\boldsymbol{A}^s}{s} = \boldsymbol{A}^0 - \boldsymbol{A}^1$, which allows simplifying the loss to

$$\mathcal{L}_{\mathrm{FM}}(\boldsymbol{\psi}) = \mathbb{E}_{s\sim\mathcal{U}([0,1]),(\boldsymbol{A}^0,\boldsymbol{O})\sim\mathcal{D},\boldsymbol{A}^1\sim\mathcal{N}(\mathbf{0},\boldsymbol{I})} \left[\left\|\boldsymbol{v}_{\boldsymbol{\psi}}\left(\boldsymbol{A}^s, \boldsymbol{O}, s\right) - \left(\boldsymbol{A}^0 - \boldsymbol{A}^1\right)\right\|_2\right]. \quad (38)$$

After training, we can generate a new action chunk $\boldsymbol{A}^0 \sim q(\cdot|\boldsymbol{O})$ by sampling $\boldsymbol{A}^1 \sim \mathcal{N}(\mathbf{0}, \boldsymbol{I})$ and numerically integrating the ODE with the learned velocity field, for example, through a simple Euler integration

$$\boldsymbol{A}^{s-\delta s} = \boldsymbol{A}^s + \delta s \boldsymbol{v}_{\boldsymbol{\psi}}\left(\boldsymbol{A}^s, \boldsymbol{O}, s\right) \quad (39)$$

with $\lceil 1/\delta s\rceil$ steps.

### B.4 Metrics

We denote the number of failed rollouts (positives) by #P and the number of successful rollouts (negatives) by #N. The number of true positives, i.e., failed rollouts correctly flagged as `Fail` by the failure predictor, is denoted by #TP. Similarly, #FP, #TN, and #FN are the numbers of false positives, true negatives, and false negatives, respectively. It holds that $\#\mathrm{TP} + \#\mathrm{FN} = \#P$ and $\#\mathrm{FP} + \#\mathrm{TN} = \#N$. The accuracy and balanced accuracy are given by

$$\mathrm{ACCURACY} = \frac{\#\mathrm{TP} + \#\mathrm{TN}}{\#P + \#N},$$

$$\mathrm{BALANCED\ ACCURACY} = \frac{1}{2}\left(\frac{\#\mathrm{TP}}{\#P} + \frac{\#\mathrm{TN}}{\#N}\right).$$

We introduce a novel metric to measure the two desired capabilities of a failure predictor, raising a failure warning correctly *and* early, with a single scalar. Denoting the detection times of true positives by $t_i$, $i = 1, \ldots, \#\mathrm{TP}$, we define the (balanced) timestep-wise-accuracy (TWA) as

$$\mathrm{TWA} = \frac{\sum_i(1 - t_i/T) + \#\mathrm{TN}}{\#P + \#N}, \quad (40)$$

$$\mathrm{BALANCED\ TWA} = \frac{1}{2}\left(\frac{\sum_i(1 - t_i/T)}{\#P} + \frac{\#\mathrm{TN}}{\#N}\right). \quad (41)$$

Our idea behind this metric (and the reason for its name) is as follows: When considering a failed episode of length $T$ which is correctly flagged as `Fail` at time $t$, the failure predictor is correct about the final rollout outcome only at timesteps $t, \ldots, T$. Thus, the proportion of timesteps for which the prediction is correct is given by $\frac{T-t}{T} = 1 - \frac{t}{T}$.

To avoid bias from imbalanced datasets with an unequal number of failed and successful rollouts, we report the balanced accuracy and balanced TWA throughout this work, omitting "balanced" for brevity.

### B.5 Baselines

**PCA-kmeans:** Liu et al. [42] train a failure detector on image embeddings of a visual world model, assuming the availability of failure trajectories. Using Principal Component Analysis (PCA) to reduce the embedding dimension and k-means clustering to calculate 64 centroids, failure situations are detected based on the distance to the closest centroid. For our experiments, we reimplement the method for the policy embedding space and compute the clusters from successful ID rollouts.

**logpZO:** Xu et al. [75] propose a failure detection approach based on learning the distribution of observation embeddings via flow matching. For a new observation $O_t$, the ODE is solved backwards starting from $O_t^{\mathrm{e}}$ via the learned velocity field, yielding a latent noise sample $Z_{O_t}$. The dissimilarity score is determined from the likelihood of $Z_{O_t}$ under the Gaussian source distribution and given by $\|Z_{O_t}\|_2^2$. We reimplement this approach as a baseline, adopting a U-Net architecture as proposed by the authors.

**STAC:** Agia et al. [1] propose measuring the cumulative statistical temporal action consistency (STAC) to detect policy failures. STAC calculates the divergence between the temporally overlapping components of the action distributions at two consecutive policy timesteps using the maximum mean discrepancy (MMD) with a radial basis function (RBF) kernel. We directly adopt STAC as a baseline, using some of the code released by the authors under an MIT License. Through our experiments, we observe that STAC struggles with observation-dependent action multimodality, a characteristic that most of our tasks exhibit. The reason for this is that the temporally overlapping parts of the action sequence batches directly before and after committing to a behavior mode are often very different because their generation is conditioned on *different* observations, as visualized in Figure 3 d).

**RND-A:** He et al. [22] use RND trained on trajectories to learn a confidence score that measures the reliability of generated actions for successful task completion. We adapt this approach to action chunks, adopting some of the code released by the authors under an MIT License, and calibrate the resulting uncertainty scores to predict policy failures.

Several recent works have proposed using VLMs to externally monitor a robot and reason about erroneous behavior [1, 17]. However, because these methods lack information about the policy's input and output distribution, they are inherently unsuitable for predicting failures early before they occur, which is one of the main objectives of this work. For this reason, we do not use VLM-based approaches as baselines.

### B.6 Compute Resources

We conduct all experiments on a workstation with 64 GB of RAM, an NVIDIA GeForce RTX 4090 GPU, and an Intel Core i9-285 K CPU. Evaluating FIPER and the baselines for all tasks, window sizes, and quantile values takes about one hour per seed. We note that computing the ACE score and running generative (vision-based) policies could be challenging on some mobile platforms with limited computational resources.

## C Extended Results

### C.1 Uncertainty Scores

As a supplement to Figure 4, we show the distribution of uncertainty scores for the four rollout categories Success ID, Success OOD, Fail ID, and Fail OOD, in Figure 9. Here, we consider only

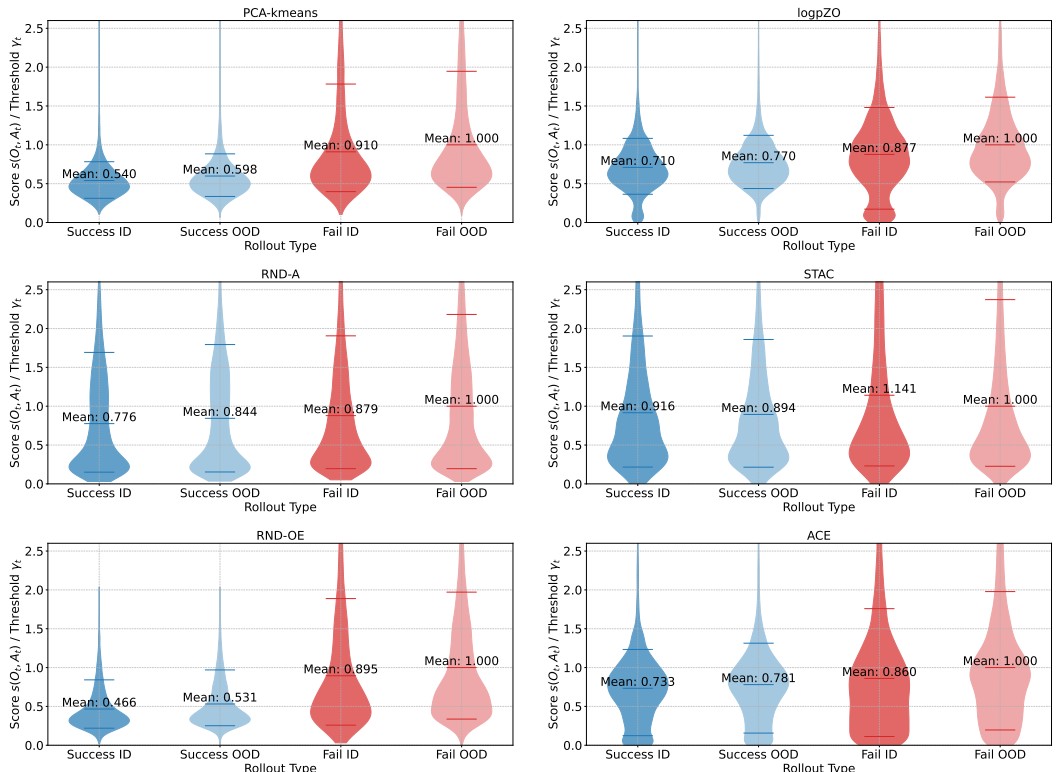

Figure 9: Distribution of the uncertainty scores $s(\boldsymbol{O}_t, \boldsymbol{A}_t)$, normalized by the corresponding CP band threshold $\gamma_t$, for different types of rollouts. We divide all values by the mean for Fail OOD for better comparability. The top and bottom horizontal lines correspond to the 90% and 10% quantiles, respectively.

the three simulation tasks, as there is no distinction between ID and OOD rollouts for the real-world tasks (see Table 4). RND-OE shows a clearer distinction between successful OOD rollouts and failed ID rollouts than the other observation-based scores, PCA-kmeans and logpZO, mainly because high RND-OE scores occur much more frequently in failure rollouts than in successful rollouts, regardless of ID or OOD. The data for the ACE score has outliers because the time-varying threshold sometimes takes very small values, which makes it difficult to visually interpret the distribution. Nonetheless, high values occur much more frequently for Fail ID than Success OOD rollouts, and our action-based failure predictor alone achieves the second-highest accuracy, which underlines the strengths of ACE to distinguish failed from successful rollouts. Moreover, our proposed approach of aggregating uncertainty scores over a sliding window increases the robustness against such outliers in the calibration rollouts.

## C.2 Extended Baseline Comparison

Table 6 is an extended version of Table 1, additionally including the TPR and TNR. To avoid selecting a window size $w$ that is more beneficial for some methods but less so for others, we use the value of $w$ that yields the highest TWA across all tasks. This "best" window size is also reported in Table 6. The results show that no baseline achieves a TPR *and* TNR greater than 0.4 for all environments, while the lowest value with FIPER is a TNR of 0.44 for the PUSHT environment. This highlights that FIPER can predict failures *and* recognize OOD situations that lead to success more robustly than existing approaches. Compared to our two primary baselines, STAC and logpZO, which are both designed specifically for failure detection, FIPER achieves a notably higher overall TPR, while at the same time not suffering from a low TNR like PCA-kmeans. Importantly, FIPER's strong performance is evident across a diverse set of environments, rather than being limited to a particular type of robot embodiment or task.

Table 6: We compare FIPER and its individual failure predictors (3) and (6) to four state-of-the-art baselines for OOD and failure detection. We highlight the best and underline the second-best value in each row. Detection times (DT) in brackets have corresponding TPR or TNR below 0.4, both of which indicate poor discrimination between successes and failures. We report the mean and standard deviation across five seeds. This table is an extension of Table 1.

| Task | Metric | PCA-kmeans [42] | logpZO [75] | RND-A [22] | STAC [1] | RND-OE | ACE | **FIPER** |
|---|---|---|---|---|---|---|---|---|
| Best $w$
Best threshold | | 40
time-var. | 15
CP constant | 2
CP constant | 15
CP constant | 1
CP constant | 45
time-var. | 25/50
time-var. |
| Sorting | TWA ↑ | $0.49^{\pm0.00}$ | $\underline{0.54}^{\pm0.00}$ | $\mathbf{0.55}^{\pm0.01}$ | $0.46^{\pm0.00}$ | $0.52^{\pm0.01}$ | $\underline{0.54}^{\pm0.00}$ | $\underline{0.54}^{\pm0.00}$ |
| | Acc. ↑ | $0.56^{\pm0.00}$ | $\mathbf{0.67}^{\pm0.00}$ | $0.62^{\pm0.01}$ | $0.48^{\pm0.00}$ | $0.59^{\pm0.02}$ | $0.65^{\pm0.00}$ | $\underline{0.66}^{\pm0.00}$ |
| | DT ↓ | $(0.15^{\pm0.00})$ | $0.48^{\pm0.01}$ | $0.34^{\pm0.03}$ | $(0.46^{\pm0.00})$ | $(0.17^{\pm0.03})$ | $\mathbf{0.30}^{\pm0.00}$ | $\underline{0.32}^{\pm0.00}$ |
| | TPR ↑ | $0.93^{\pm0.00}$ | $0.57^{\pm0.01}$ | $0.43^{\pm0.09}$ | $0.09^{\pm0.00}$ | $0.92^{\pm0.03}$ | $0.76^{\pm0.00}$ | $0.75^{\pm0.00}$ |
| | TNR ↑ | $0.20^{\pm0.00}$ | $0.77^{\pm0.01}$ | $0.81^{\pm0.08}$ | $0.86^{\pm0.00}$ | $0.26^{\pm0.06}$ | $0.54^{\pm0.00}$ | $0.57^{\pm0.00}$ |
| Stacking | TWA ↑ | $\mathbf{0.66}^{\pm0.00}$ | $0.58^{\pm0.00}$ | $0.50^{\pm0.00}$ | $0.59^{\pm0.00}$ | $\underline{0.62}^{\pm0.04}$ | $\underline{0.62}^{\pm0.00}$ | $\underline{0.62}^{\pm0.00}$ |
| | Acc. ↑ | $\mathbf{0.75}^{\pm0.00}$ | $0.69^{\pm0.01}$ | $0.56^{\pm0.01}$ | $0.66^{\pm0.00}$ | $0.70^{\pm0.06}$ | $\underline{0.73}^{\pm0.00}$ | $\underline{0.73}^{\pm0.00}$ |
| | DT ↓ | $\underline{0.19}^{\pm0.00}$ | $0.49^{\pm0.00}$ | $(0.44^{\pm0.01})$ | $(0.38^{\pm0.00})$ | $\mathbf{0.16}^{\pm0.05}$ | $0.27^{\pm0.00}$ | $0.28^{\pm0.00}$ |
| | TPR ↑ | $1.00^{\pm0.00}$ | $0.48^{\pm0.02}$ | $0.27^{\pm0.04}$ | $0.35^{\pm0.00}$ | $0.98^{\pm0.02}$ | $0.84^{\pm0.00}$ | $0.84^{\pm0.00}$ |
| | TNR ↑ | $0.51^{\pm0.00}$ | $0.90^{\pm0.00}$ | $0.85^{\pm0.02}$ | $0.96^{\pm0.00}$ | $0.42^{\pm0.14}$ | $0.61^{\pm0.00}$ | $0.62^{\pm0.00}$ |
| PushT | TWA ↑ | $0.53^{\pm0.00}$ | $0.52^{\pm0.00}$ | $0.52^{\pm0.01}$ | $\mathbf{0.58}^{\pm0.00}$ | $0.54^{\pm0.01}$ | $\underline{0.56}^{\pm0.00}$ | $0.55^{\pm0.00}$ |
| | Acc. ↑ | $0.58^{\pm0.00}$ | $0.55^{\pm0.00}$ | $0.55^{\pm0.01}$ | $0.71^{\pm0.00}$ | $0.55^{\pm0.01}$ | $\underline{0.71}^{\pm0.00}$ | $\mathbf{0.71}^{\pm0.00}$ |
| | DT ↓ | $(0.11^{\pm0.00})$ | $(0.26^{\pm0.00})$ | $(0.20^{\pm0.02})$ | $0.52^{\pm0.00}$ | $(0.02^{\pm0.00})$ | $\mathbf{0.31}^{\pm0.00}$ | $\underline{0.32}^{\pm0.00}$ |
| | TPR ↑ | $1.00^{\pm0.00}$ | $0.17^{\pm0.00}$ | $0.31^{\pm0.03}$ | $0.49^{\pm0.00}$ | $0.80^{\pm0.04}$ | $0.99^{\pm0.00}$ | $0.98^{\pm0.00}$ |
| | TNR ↑ | $0.17^{\pm0.00}$ | $0.93^{\pm0.00}$ | $0.80^{\pm0.02}$ | $0.93^{\pm0.00}$ | $0.31^{\pm0.06}$ | $0.43^{\pm0.00}$ | $0.44^{\pm0.00}$ |
| Pretzel | TWA ↑ | $0.64^{\pm0.00}$ | $0.58^{\pm0.03}$ | $0.51^{\pm0.05}$ | $0.51^{\pm0.00}$ | $0.55^{\pm0.02}$ | $\mathbf{0.75}^{\pm0.00}$ | $\underline{0.68}^{\pm0.03}$ |
| | Acc. ↑ | $0.65^{\pm0.00}$ | $0.65^{\pm0.04}$ | $0.53^{\pm0.05}$ | $0.67^{\pm0.00}$ | $0.72^{\pm0.02}$ | $\underline{0.82}^{\pm0.00}$ | $\mathbf{0.85}^{\pm0.00}$ |
| | DT ↓ | $(0.01^{\pm0.00})$ | $\underline{0.24}^{\pm0.04}$ | $(0.52^{\pm0.36})$ | $0.44^{\pm0.00}$ | $0.44^{\pm0.02}$ | $\mathbf{0.13}^{\pm0.00}$ | $0.33^{\pm0.07}$ |
| | TPR ↑ | $1.00^{\pm0.00}$ | $0.54^{\pm0.05}$ | $0.13^{\pm0.12}$ | $0.69^{\pm0.00}$ | $0.77^{\pm0.12}$ | $1.00^{\pm0.00}$ | $1.00^{\pm0.00}$ |
| | TNR ↑ | $0.30^{\pm0.00}$ | $0.76^{\pm0.05}$ | $0.92^{\pm0.06}$ | $0.64^{\pm0.00}$ | $0.67^{\pm0.09}$ | $0.64^{\pm0.00}$ | $0.70^{\pm0.00}$ |
| PushChair | TWA ↑ | $0.50^{\pm0.00}$ | $\underline{0.78}^{\pm0.02}$ | $0.71^{\pm0.05}$ | $0.73^{\pm0.00}$ | $0.74^{\pm0.11}$ | $0.69^{\pm0.00}$ | $\mathbf{0.83}^{\pm0.02}$ |
| | Acc. ↑ | $0.50^{\pm0.00}$ | $\underline{0.92}^{\pm0.02}$ | $0.82^{\pm0.05}$ | $0.88^{\pm0.00}$ | $0.80^{\pm0.14}$ | $0.80^{\pm0.00}$ | $\mathbf{0.96}^{\pm0.02}$ |
| | DT ↓ | $(0.00^{\pm0.00})$ | $0.26^{\pm0.01}$ | $\underline{0.21}^{\pm0.02}$ | $0.30^{\pm0.00}$ | $\mathbf{0.11}^{\pm0.07}$ | $0.23^{\pm0.00}$ | $0.27^{\pm0.00}$ |
| | TPR ↑ | $1.00^{\pm0.00}$ | $1.00^{\pm0.00}$ | $1.00^{\pm0.00}$ | $1.00^{\pm0.00}$ | $0.98^{\pm0.03}$ | $1.00^{\pm0.00}$ | $1.00^{\pm0.00}$ |
| | TNR ↑ | $0.00^{\pm0.00}$ | $0.83^{\pm0.03}$ | $0.64^{\pm0.10}$ | $0.75^{\pm0.00}$ | $0.61^{\pm0.30}$ | $0.60^{\pm0.00}$ | $0.93^{\pm0.04}$ |
| Average | TWA ↑ | $0.57^{\pm0.00}$ | $0.60^{\pm0.01}$ | $0.56^{\pm0.02}$ | $0.57^{\pm0.00}$ | $0.59^{\pm0.04}$ | $\underline{0.63}^{\pm0.00}$ | $\mathbf{0.65}^{\pm0.01}$ |
| | Acc. ↑ | $0.61^{\pm0.00}$ | $0.69^{\pm0.01}$ | $0.62^{\pm0.03}$ | $0.68^{\pm0.00}$ | $0.67^{\pm0.05}$ | $\underline{0.74}^{\pm0.00}$ | $\mathbf{0.78}^{\pm0.00}$ |
| | DT ↓ | $(0.09^{\pm0.00})$ | $0.35^{\pm0.01}$ | $0.34^{\pm0.09}$ | $0.42^{\pm0.00}$ | $\mathbf{0.18}^{\pm0.03}$ | $\underline{0.25}^{\pm0.00}$ | $0.30^{\pm0.02}$ |
| | TPR ↑ | $0.99^{\pm0.00}$ | $0.55^{\pm0.02}$ | $0.43^{\pm0.06}$ | $0.52^{\pm0.00}$ | $0.89^{\pm0.05}$ | $0.92^{\pm0.00}$ | $0.92^{\pm0.00}$ |
| | TNR ↑ | $0.24^{\pm0.00}$ | $0.84^{\pm0.02}$ | $0.80^{\pm0.06}$ | $0.83^{\pm0.00}$ | $0.45^{\pm0.13}$ | $0.56^{\pm0.00}$ | $0.65^{\pm0.01}$ |

## C.3 Impact of the Window Size for Score Aggregation

Complementary to our discussion in Section 6, we provide more detailed results about our proposed approach of aggregating uncertainty scores over a sliding window of length $w$. Figure 10 shows the impact of $w$ for a different threshold definitions, averaged across quantiles $1 - \delta \in \{0.9, 0.91, \ldots, 0.99\}$. For FIPER, we set $w_O = w_A = w$ in this analysis for simplicity. Using $w = 1$, i.e., considering only the current observation and/or action [75], leads to low failure prediction accuracy. Increasing the window size improves accuracy, but mostly at the expense of slower detection. In general, we observe that the optimal window size with respect to TWA varies significantly across different failure prediction methods and threshold types. This supports our approach of reporting the main results in Tables 1 and 6 for the window size that achieves the highest TWA across all tasks for the respective method.

It mainly depends on the task whether a decrease in DT justifies a drop in accuracy. In domains where rapid response to imminent failures is crucial, such as autonomous surgical assistance or collaborative assembly lines, earlier failure warnings may be highly desirable, even if they mean more false alarms. Conversely, in a logistics scenario, misplacement of objects is often not dangerous, whereas every incorrect failure warning would have to be checked manually by a human and would thus be costly. In this case, a more conservative aggregation over a longer window may better balance operational efficiency and failure prevention.

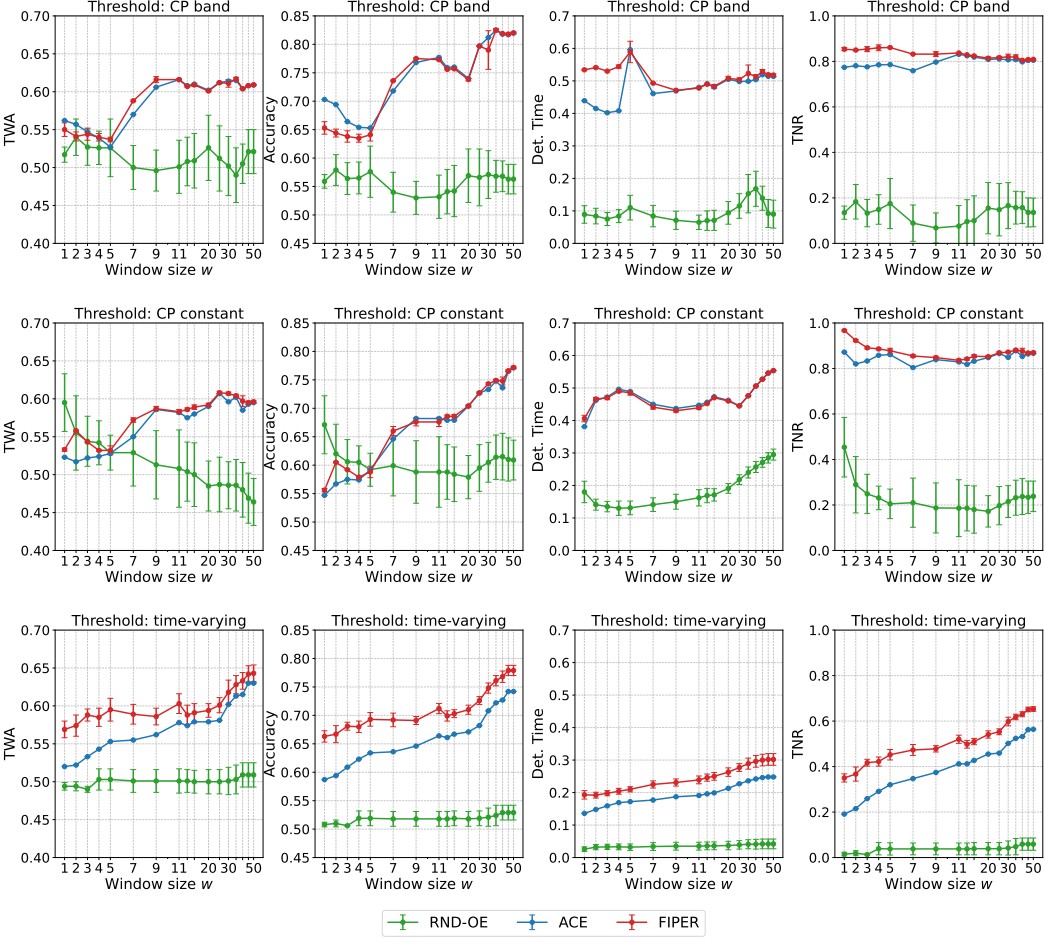

Figure 10: Impact of the sliding window size $w$ for three threshold types. The results are averaged across five seeds, with the bars indicating the standard deviation.

## C.4 Impact of the Threshold Computation

The performance of a failure predictor is affected by the definition of its threshold, in particular, by (i) the type of threshold and (ii) the design parameter $\delta$ for calibration, which controls the desired FPR. We have conducted multiple ablation studies to analyze the impact of these factors.

### C.4.1 Quantiles

We average our results over a set of quantiles $1 - \delta \in \{0.9, 0.91, \ldots, 0.99\}$, arguing that there is no "best" quantile value for calibration, and because we want to avoid cherry-picking a specific quantile that is more favorable to the performance of one method than others. In Figure 11, we show the impact of different quantile values $1 - \delta$ on TWA, DT, TPR, and TNR, using an exemplary window size of $w = 15$ for all methods. We observe that $1 - \delta$ directly affects both TPR and TNR. This is expected, as the quantile value effectively controls the threshold at which a rollout is flagged as Fail. Consequently, increasing $1 - \delta$ leads to fewer predicted failures, which decreases TPR and increases TNR. These two effects largely cancel out in terms of overall accuracy and TWA, leading us to the above claim that there is no "best" quantile.

**False Positive Rate (FPR)** As discussed in Appendix A.3, the hyperparameter $\delta \in (0, 1)$ effectively controls the FPR on the calibration rollouts. As shown in Figure 11, the CP constant threshold and one-sided CP band defined in Propositions 2 and 3 lead to very high TNR but fairly low TPR. In contrast, our simpler time-varying threshold yields a much higher TPR at the cost of raising false

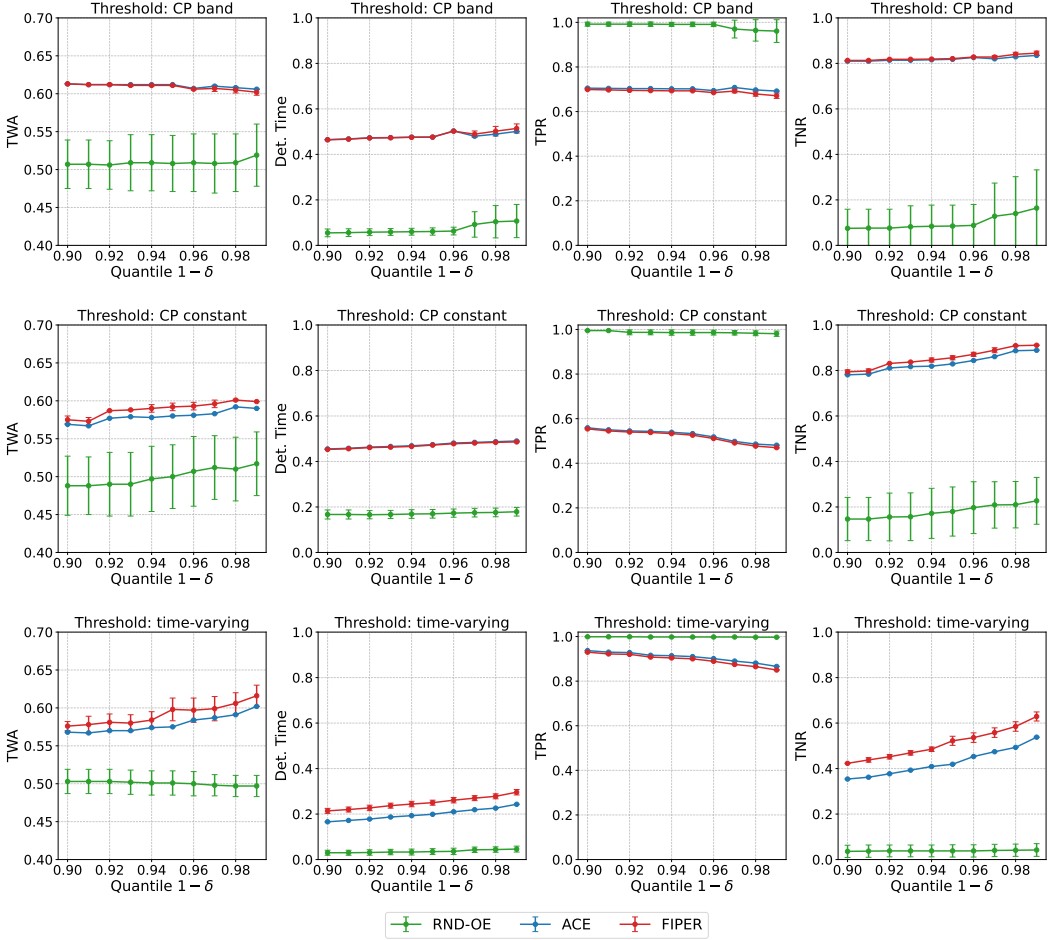

Figure 11: Impact of the quantile value $1 - \delta$ used for calibration for three threshold types. The results are reported for a sliding window size of $w = 15$ and averaged across five seeds, with the bars indicating the standard deviation.

alarms more frequently. Which threshold type is most suitable, therefore, depends on whether recognizing all failures or avoiding false alarms has higher priority. The fact that neither a constant threshold nor a one-sided CP band achieves the desired FPR exactly matches the results reported in prior works [1, 75] and stems from the fact that Propositions 2 and 3 only hold for ID rollouts, whereas we also evaluate on OOD scenarios. We hypothesize that strictly separating the rollouts used for training the failure predictors from those used for calibration, as well as using a larger calibration dataset, could mitigate this issue.

### C.4.2 Threshold Type

For our main results in Tables 1 and 6, we report the results with the best-performing threshold definition (CP band, CP constant, or time-varying) for each method. While no single threshold type consistently outperforms the others across all methods, we perform a direct comparison in Figure 12 and analyze the results below.

**CP-based thresholds detect rather than predict failures.** Since the CP band and CP constant thresholds are defined such that the probability of raising a false alarm is upper-bounded, their values $\gamma_t$ are higher than those of the time-varying threshold that is calculated timestep-wise. Consequently, the CP-based thresholds yield a much higher TNR but also predict failures much later. FIPER achieves the highest TWA values overall with our time-varying threshold and a larger window size. While accuracy may be more important than early detection in some cases, we argue that

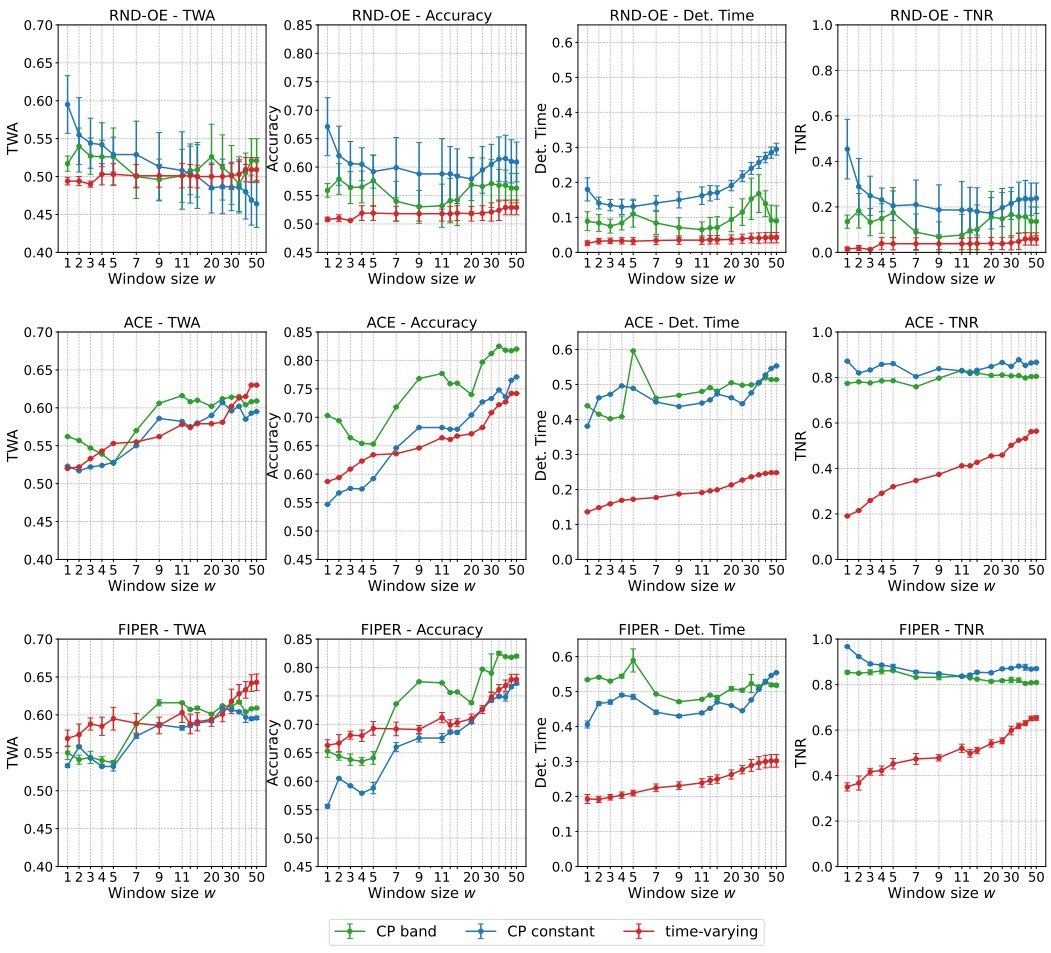

Figure 12: Impact of the threshold type used by the failure predictors. The results are averaged across five seeds, with the bars indicating the standard deviation.

the conservative nature of CP-based thresholds is more suitable for failure detection rather than prediction.

**Constant thresholds are less accurate for FIPER**   The CP band and time-varying thresholds yield higher TWA and accuracy for FIPER. We attribute this to the fact that these threshold types effectively compare the failure prediction score at timestep $t$ against the scores in the calibration dataset for the *same* timestep. Consequently, they are more sensitive to temporal anomalies in the score values, which can stem, for example, from the robot failing to grasp an object. However, the higher TNR of the constant threshold shows that the CP band and time-varying threshold raise false alarms more frequently in generalization scenarios, where the policy may solve subtasks in varying order or with inconsistent timing.

### C.5   Discussion of Action-Based Scores

We briefly compare our action-based failure predictor, ACE, specifically against the two baselines that also operate on the policy output, RND-A and STAC. The results in Table 6 demonstrate that ACE achieves higher TWA and accuracy and lower DT than RND-A and STAC. This demonstrates the superiority of our proposed ACE score for quantifying task-relevant uncertainty from the generated actions. STAC [1], a recent baseline, notably achieves a TPR below 0.5 in the SORTING, STACKING, and PUSHT environments that are characterized by strong action multimodality. In comparison, our entropy-based score can predict between 76% and 99% of all failures for these tasks. ACE is specifically designed to capture the uncertainty inherent in multimodal action distributions, and

it outperforms STAC's temporal consistency score in our experiments. Figure 3 illustrates the conceptual advantage of measuring entropy instead of temporal consistency in the action distributions in tasks that involve action multimodality.

### C.6 Discussion of Observation-Based Scores

RND-OE yields a clearer separation between successful and failed rollouts than the observation-based baselines, PCA-kmeans and logpZO, as shown in Figures 4 and 9. Moreover, RND-OE can predict failures more rapidly and achieves the lowest DT overall. This can be especially beneficial in scenarios involving humans, where high sensitivity to different kinds of failures is crucial. In comparison, PCA-kmeans suffers from a low overall TNR, effectively labeling most rollouts as failures (see Table 6). Although logpZO can match the accuracy and TWA of RND-OE (Table 6), our RND-OE score achieves much lower DT, highlighting its advantages for predicting failures before they occur. We find that the design of FIPER benefits from the high sensitivity of RND-OE because a rollout is only flagged as `Fail` if there is also high action uncertainty.

We also considered RND models that take the raw observation images rather than embeddings as input, which is closer to the original version of RND proposed by Burda et al. [9]. However, operating directly in the embedding space of the policy performed significantly better, and we argue that this approach offers two key advantages: First, because the observation encoder is trained end-to-end with the policy, the embeddings naturally filter out details in the observations that are irrelevant to the policy and, thus, for reasoning about task failure. Conversely, performing OOD detection directly on raw images can yield a high FPR in real-world tasks, as the RND model cannot separate visual noise from policy-relevant features that are indicative of failure. Second, since all common IL policies have an observation encoding module, directly using this pre-trained feature extractor generally allows us to RND-OE from a smaller rollout dataset, reducing data collection effort.

## D Limitations

This section summarizes what we consider to be the main limitations of this work.

**The failure prediction accuracy might still be insufficient for certain applications.** Although FIPER outperforms state-of-the-art baselines across various tasks, its overall failure prediction accuracy for the best TWA is only 78%, highlighting the difficulty of early failure prediction in the absence of failure data. This accuracy may still be too low for certain applications, such as a robot operating in an assembly line, where true policy failures need to be recognized early and reliably, but false alarms are costly. Possible ways to improve the failure prediction accuracy could be increasing the number of calibration rollouts or using the training data of the policy.

**The inclusion of historical data could be further improved.** FIPER considers historical data solely through the aggregation of past uncertainty scores using a sliding window, without incorporating this historical context into the actual calculation of uncertainty scores. We hypothesize that the history of previous observations and action predictions could contain early warning signs indicative of policy failures, potentially enhancing the ability of FIPER to anticipate them.

**RND-OE needs to be trained.** We outline the benefits of using RND-OE compared to other observation-embedding-based methods in Appendix C.6. However, having to train an RND-OE model that is separate from the policy is still a limitation of our approach.

**We only test RND-OE with image-based observation embeddings.** In our experiments, we only use vision-based IL policies as they do not require specialized sensors. Therefore, although we expect FIPER to work well in principle for additional input modalities, such as language, touch or audio, this remains an untested hypothesis for the time being.

**Time-varying thresholds can be restrictive.** Time-varying thresholds can be disadvantageous for two reasons. First, they implicitly assume that the temporal sequence of events in a successful rollout is always similar. However, this may not be the case, for example, if the policy manages to grasp an object only in the second attempt, temporally shifting the trajectory from its expected pattern.

Therefore, if the training data contains multiple temporally distinct ways of completing the task, a constant threshold may be more suitable. Second, a time-varying threshold can only be calculated for timesteps that are represented sufficiently often in the calibration dataset. This can be a problem if rollouts have different lengths and may require padding.

**Aleatoric uncertainty could impact ACE.**   If there is high variability (aleatoric uncertainty) in the demonstration data, this is generally reflected by the learned action distribution of the policy. In this case, the variability within generated action chunk batches can be high even for successful calibration rollouts, potentially leading to a larger threshold, which may result in a lower TPR. We think that disentangling aleatoric from epistemic uncertainty for generative policies remains an interesting avenue for future research.

