# OpenReview forum: "Failure Prediction at Runtime for Generative Robot Policies"
_NeurIPS.cc/2025/Conference — NeurIPS 2025 poster_

### Official Review · Reviewer_a998 · 2025-06-04

**Clarity:** 3
**Significance:** 3
**Originality:** 3
**Rating:** 5
**Confidence:** 3

**Summary:**

The paper introduces FIPER, a framework for predicting failures at runtime in generative imitation learning policies without requiring failure examples. Existing failure detection approaches either trigger false alarms (via out-of-distribution detection) or react too late (via external monitoring). FIPER addresses these limitations by combining two signals: sequences of out-of-distribution observations and sustained high action uncertainty. These signals are calibrated on successful rollouts, and conformal prediction is applied to offer statistical guarantees. The method is evaluated across five simulated and real-world domains, demonstrating earlier and more accurate failure prediction than baselines. This work enhances the safety and interpretability of deploying generative IL policies in real-world robotics.

**Questions:**

1. Could you elaborate on Equation 4? Specifically, how is the estimated entropy computed, and why is this approach considered more efficient? These details are essential to support your method but are unclear in the main text. (Note: I later found the explanation in the appendix—consider adding a reference in the main paper for clarity.)

2. In Equation 7, have you explored alternative ways to combine the two criteria, such as using weighted combinations? It would be helpful to understand whether other formulations were considered.

3. How is the threshold δ chosen, and how does its variation impact performance? Some discussion or sensitivity analysis would strengthen the empirical justification.

4. Why do you consider different generative modeling techniques and policy backbones for different tasks?

**Ethical Concerns:**

["NO or VERY MINOR ethics concerns only"]

**Final Justification:**

From my perspective, this work is practical and holds value for many real-world applications. However, in its original submission, it missed some key points. My initial concerns focused primarily on:

- Improving the clarity and readability of the paper

- Refining experimental details and ablation studies to support plausibility

In the authors' rebuttal, these concerns were thoughtfully addressed. I’m pleased to see that the planned revisions for the camera-ready version incorporate these improvements. Based on the resolved issues and the authors’ responsiveness, I will maintain my original score recommendation.

**Limitations:**

yes

**Quality:**

3

**Strengths And Weaknesses:**

Strengths:

1. The paper is clearly written and easy to follow.
2. The proposed method is theoretically well-grounded and clearly explained.
3. The method demonstrates strong empirical performance.
4. The authors conduct extensive experiments across diverse settings.

Weaknesses:
The paper's organization could be improved by more clearly referencing the appendix. Much of the theoretical content is deferred there, leaving some explanations in the main text vague.

---

> ### Author Rebuttal · Authors · 2025-07-30
>
> We are grateful for your comments and feedback and appreciate your recognition of our method's performance. We have responded to your questions below:
> - **References to the Appendix:** As you suggested, we have included a reference to the detailed explanation of the entropy computation in Appendix A.1 below Equation 4. Moreover, we have added references to the other methodological and theoretical details in Appendix A.2, A.3, and A.4 to the corresponding passages in Section 4 and believe these changes significantly improve the organization of the methodology section.
> - **Alternative ways of combining detector outputs:** Besides logically combining the detector outputs, we have also tried a weighted combination of the normalized scores (i.e., the score at timestep $t$ divided by the corresponding threshold). This approach would allow selecting the influence of the individual scores in the final output, but the weighting factor would represent an additional (task-specific) hyperparameter. We wanted to avoid further increasing the number of hyperparameters to ensure generalizability of our results across tasks and environments, and to not increase the amount of tuning required. For this reason, and because the weighted combination did not perform better in our experiments, we have not included this variant in our final results. We added this information in two sentences after line 742 in Appendix A.4. Lastly, we would like to highlight that we compare the logical AND combination of the detector outputs to an OR combination in Table 2 and find the latter to be a viable alternative when aiming primarily for high sensitivity and low detection time.
> - **Choice and impact of $\\delta$:** The value of $\delta$, or equivalently the quantile value $1-\\delta$, is a crucial parameter, since the score thresholds are defined using conformal prediction such that the false-positive-rate on the calibration rollouts is $\\leq \\delta$. Due to space limitations in the main paper, the analysis of the impact of $1-\\delta$ is included in Appendix C.4.2 in Figure 11. The results show that a lower value of $1-\\delta$ increases the sensitivity to failures but also increases the rate of false alarms. It may depend on the task whether false negatives or false positives are more costly, and the TWA plots on the left of Figure 11 show there is no single "best" value for $\\delta$ when aiming for both accurate and early failure prediction. Therefore, and to conduct a fair comparison of the different methods instead of cherry-picking $\\delta$, we report all our main results averaged across $1-\\delta \in \\{0.9,0.91,...,0.99\\}$. To better justify this evaluation strategy in the main paper, we have added a reference to Appendix C.4.2 to the description of our evaluation protocol in line 273.
> - **Policy design:** In our experiments, we wanted to test our method for the two most popular generative modeling techniques (flow matching and diffusion) and policy backbones (transformers and CNNs). We chose to use flow matching with a transformer backbone for two of the tasks and diffusion with a CNN backbone for the other three tasks (instead of evaluating all possible combinations of generative models and backbones for each task) to keep the computational effort feasible for us. For the PushT and PushChair tasks, we use the datasets from prior work [1], so the choice of using a CNN-based diffusion policy for these tasks had already been made by those authors. To clarify these points, we have expanded the paragraph in lines 794 to 798 in Appendix B.2 and, in particular, added the following:
>
>   *"We combine flow matching with a transformer backbone for the Sorting and Stacking tasks and combine diffusion with a CNN backbone for the other three tasks. This choice was made to evaluate our method's performance for different policy formulations and architectures, and we have made it independently of the tasks' characteristics."*
>
> We hope this response addresses your questions. Please let us know if you would like us to provide further details.

---

> > ### Comment · Reviewer_a998 · 2025-08-03
> >
> > The authors' rebuttal has addressed my concerns. I will maintain my original rating.

---

### Official Review · Reviewer_Hshp · 2025-07-02

**Clarity:** 3
**Significance:** 4
**Originality:** 2
**Rating:** 4
**Confidence:** 4

**Summary:**

This paper proposes FIPER, a framework for predicting task failures of generative robot policies at runtime without requiring any failure data. FIPER identifies two key indicators of impending failure: 1) consecutive OOD observations in the policy's embedding space and 2) persistently high entropy in the generated action distributions. These scores are aggregated over short time windows and statistically calibrated using conformal prediction on a small set of successful rollouts. FIPER raises an alarm only when both scores exceed their calibrated thresholds simultaneously (logical AND), ensuring robustness. Evaluated across five diverse simulation and real-world robotic tasks with various generative policies, FIPER enables earlier, more accurate failure warnings, enhancing safety and interpretability for robots in human-centered environments.

**Questions:**

1. The paper treats consecutive OOD observations and persistently high action entropy as independent indicators of policy failures (Section 4.3), but their frequent co-occurrence in failures (Fig. 3) suggests a potential causal link: could OOD observations induce high-entropy actions? Specifically: Is there evidence that OOD inputs directly cause the policy’s action distribution to become high-entropy ? If not, could FIPER miss failures where these indicators decouple—e.g.,
- OOD without high entropy (policy handles novelty confidently but still fails)?
- High entropy without OOD (policy acts erratically in-distribution)?


2. In the current experiment, how are failure situations handled after the failure prediction at runtime. Can appropriate interventions be made on the robot's state after failure prediction to further improve the success rate of the task?
3. The experiments only involve Franka and manipulators. Will the same excellent performance be achieved in other tasks, such as robot locomotion tasks?

**Ethical Concerns:**

["NO or VERY MINOR ethics concerns only"]

**Final Justification:**

The authors have addressed my concerns in their rebuttal. This algorithm can identify potential risks during the operation of the policy, and in the future, combining with recovery algorithms, it will be able to achieve more extensive applications.

I adjust my score to 4.

**Limitations:**

yes

**Paper Formatting Concerns:**

No paper formatting concerns.

**Quality:**

3

**Strengths And Weaknesses:**

Strengths
1. **No Failure Data Required**:
  - Trains solely on successful demonstrations, eliminating risky data collection for failure scenarios.
2. **Dual-Signal Detection**:
  - Combines temporal OOD observations (via RND-OE) and persistent action uncertainty (via ACE), reducing false alarms and enabling early failure prediction.


Weaknesses
1. **Action Space Scalability**:
The binning approach used in calculating the ACE score may face challenges with high-dimensional actions, though this has not been tested.
2. **Threshold Sensitivity**:
Although the paper proposes range conformal prediction to calibrate thresholds, threshold adjustment must be re-performed for different tasks. Moreover, the method presented is highly sensitive to thresholds—during the threshold calibration phase, any errors in threshold calculation caused by uneven data distribution could significantly impact the algorithm’s results.

---

> ### Author Rebuttal · Authors · 2025-07-30
>
> Thank you very much for your valuable feedback and for recognizing the significance of our work. We have responded to the mentioned weaknesses and your questions below:
> - **Action space scalability:** We agree with you that the computational cost of the ACE score, in principle, increases with the action dimension. However, we calculate the ACE score in the Cartesian space using end-effector positions (see also Appendix A.1). This can be done for any manipulation task (and even for locomotion or navigation, see "Other tasks" below), even if the policy outputs joint angles or velocities, by using forward kinematics. We use this approach not only for computational reasons but also to obtain interpretable uncertainty scores in the same 3D space in which the task is performed. High uncertainty in Cartesian space means that the policy is unsure about the correct path to take (e.g., where to grasp an object), whereas entropy in, for example, joint velocities, does not necessarily correspond to task-relevant uncertainty.
> We also experimented with computing the ACE score using other action representations, such as full 6D poses, but did not observe improved performance.
>
>   For bimanual manipulation, the ACE score can be computed for each end-effector individually with our framework, and the values are then combined by averaging or taking the maximum. In fact, we evaluated the ACE score in this way on the bimanual manipulation data from C. Agia et al., "Unpacking Failure Modes of Generative Policies: Runtime Monitoring of Consistency and Progress", 2024 [1] and observed very good performance, but we could not include these experiments in our final submission, as the dataset does not contain the observation embeddings required for our RND-OE score and the PCA-kmeans and logpZO baselines.
>
>   Due to the ACE score computation in Cartesian space, we think that our method can handle common higher-dimensional action spaces well, and deploying FIPER for humanoids or quadruped locomotion would be an interesting direction for future work. We have added a sentence to the implementation details in line 243 to better emphasize this aspect of our method in the main text:
>
>   *"We compute the ACE score in the Cartesian space of predicted end-effector positions to obtain interpretable and task-relevant action uncertainty estimates and to ensure computational efficiency."*
>
>   Additionally, we have expanded lines 679 to 681 in Appendix A.1 into a separate paragraph and included the points just discussed.
>
> - **Threshold sensitivity:** We agree with you that the need to calibrate the threshold for each task is a limitation, which our method has in common with related works [1, 72]. In our opinion, this calibration step can hardly be avoided because the learned policy itself and, thus, the distribution of the failure prediction scores, depends on the underlying task-specific data distribution. However, we have observed that constant thresholds are less sensitive to subtask sequence shifts compared to time-varying thresholds. We have emphasized this insight more clearly in the extended discussion on limitations in Appendix D by adding the following to line 1027:
>
>   *"Therefore, if the training data contains multiple temporally distinct ways of completing the task, we consider a constant threshold to be more suitable."*
>
> - **Dependence of OOD observations and high action entropy:** As the action generation is conditioned on the observation, the occurrences of OOD observations and persistently high action entropy are generally not independent, and we do not make this assumption. Rather, our approach is based on the following insight: Failures of imitation learning policies represent a deviation from the distribution of successful demonstrations, which can be caused by unseen initial conditions or compounding errors, potentially due to the stochasticity of generative policies. However, many OOD situations do not result in task failure because the policy can recover from them, which is shown by the low true-negative-rate (TNR) of 50\% (see Table 6) of using our observation-based RND-OE score alone. For this reason, FIPER flags a rollout as Fail only if there is *additionally* high action entropy, which indicates the policy is also highly uncertain about the correct actions to take given the current OOD observation. This AND-combination of the two failure indications achieves the highest accuracy and can detect 87\% of all failures in our experiments (Table 6), demonstrating that most failures exhibit both indications. Using OR instead of AND can also detect all of the rare failure cases in which the two indications decouple, achieving a TPR of 100\% (Table 2), but this also means raising many false alarms. Hence, we consider the logical disjunction to be only preferable for tasks that require very high sensitivity to failures, such as assistive surgery, where every failure can have catastrophic consequences. To better highlight this aspect in the main paper, we have expanded lines 334 to 337 as follows:
>
>   *"As a variation of FIPER, the use of a logical disjunction (OR) of our two failure predictors provides a powerful alternative when the primary goal is to achieve very low DT and high TPR. This may be particularly desirable if failures could directly endanger people or cause damage to expensive items."*
>
> - **Handling of failures after prediction:** The main focus of our work is to predict policy failures accurately and early. To be able to evaluate and fairly compare different failure prediction methods, we use pre-recorded policy rollout data where we know the ground-truth final outcome (Fail or Success). However, the key motivation behind our work is to allow for intervention before a failure occurs or becomes irrecoverable. We think that carrying out such interventions during runtime after a failure warning is raised and, for example, using this data to fine-tune the policy, is a very interesting and natural direction for future work.
>
> - **Other tasks:** Our experiments involve three tasks with Franka manipulators, one mobile manipulation task with a different platform [1], and the PushT simulation task. We focus on manipulation, as we think this has recently been the largest application area of vision-based generative imitation learning policies. Nonetheless, we think that FIPER is also directly applicable to other tasks, such as navigation and locomotion, provided that two conditions are met:
>   - There is a clear definition of "task failure". This might not be as obvious for some locomotion tasks, such as tracking a desired reference base velocity, which do not have the same episodic character as typical manipulation and navigation tasks. Intuitively, "task failure" is well defined in the sense of our work if there is a maximum episode length and a clear objective that should be achieved within that time frame. For locomotion, this objective could be defined, for example, as staying within a certain tolerance around the reference speed.
>   - The actions can be represented in or transformed to Cartesian space. This is not a strict requirement, but we think it is highly beneficial for the reasons mentioned above under "Action space scalability". It applies to navigation, where the entropy can be computed for the base positions predicted from the action chunks. For locomotion, we would propose computing the action entropy for the positions of each foot individually and then combining these scores by averaging or taking the maximum, similar to bimanual manipulation (see "Action space scalability" above).
>
>   We have included this discussion in Appendix B.1.2 by adding a paragraph after the description of the real-world environments.
>
> We sincerely thank you for your insightful comments and hope our clarifications resolve your concerns. We would be happy to provide further details if needed.

---

> > ### Comment · Reviewer_Hshp · 2025-08-02
> >
> > Thank you for your rebuttal.
> > Although this paper is oriented towards generative methods, I am wondering whether similar approaches can be applied to traditional RL control tasks based on MLP?
> > In robotic tasks, due to the complexity of ACE score calculation, it may not be feasible to perform such calculations on some robot-mounted platforms with limited computing power. Nevertheless, these are engineering issues that need to be addressed in the future.
> > I will adjust the score to 4.

---

> > > ### Author Response · Authors · 2025-08-04
> > >
> > > Thank you for your comment and for increasing your rating.
> > > - We think that our approach is, in principle, also applicable to predicting failures of RL policies. The ACE score could be particularly beneficial for RL policies with multimodal action distributions, such as *"Flow Matching Policy Gradients"* by D. McAllister et al., 2025. We have added this as a potential direction for future work to the paper.
> > > - Computing the ACE score and running generative (vision-based) policies could indeed be challenging on some mobile platforms with limited computational resources. We have added a comment to Appendix B.5 "Compute Resources."

---

### Official Review · Reviewer_ZrcY · 2025-07-02

**Clarity:** 4
**Significance:** 2
**Originality:** 3
**Rating:** 4
**Confidence:** 4

**Summary:**

The paper presents a new method for failure prediction in imitation learning with generative models, called Failure Prediction at Runtime (FIPER). FIPER is based on predicting failure by looking both for consecutive out-of-distribution (OOD) observations and highly uncertain actions at the same time. OOD observations are detected with a novel method that leverages Random Network Distillation (RND), an existing reinforcement learning approach that measures observation novelty for exploration purposes. The paper also introduces timestep-wise accuracy (TWA), a new metric for evaluating failure prediction that encourages both accurate and timely predictions. FIPER is empirically evaluated on several relevant domains, generally demonstrating its capability to detect failures more accurately and faster w.r.t. the considered baselines.

**Questions:**

What is the motivation behind not conducting experiments that include humans (possibly simulated, or recorded from a dataset)? While I understand that the inclusion of these might be out of the scope of the paper, discussing what might change in these scenarios (e.g. different trade-offs between false positive rate and prediction time, which might favor different methods) would be highly beneficial.

In the discussion on the results relative to the "sliding window" experiments (306-322 and figure 4), it is not clear whether the reported gain in detection time is relevant, and whether it is enough to warrant the reported drop in accuracy. More in general, expanding the discussion with examples of when this difference might be relevant (and desirable, given the associated significant drop in accuracy) would be highly beneficial in solidifying this discussion.

If these concerns were addressed appropriately, I would raise the significance score by 1 and consider raising the overall rating.

**Ethical Concerns:**

["NO or VERY MINOR ethics concerns only"]

**Final Justification:**

The paper is sound and the authors seem to have clarified most concerns from all the reviewers.

**Limitations:**

Yes, the authors have successfully highlighted the limitations, except for those presented within the questions.

**Quality:**

3

**Strengths And Weaknesses:**

Quality: the methodology is technically sound: the relevance of the introduced metrics is empirically demonstrated, and experiments on FIPER support the claims, even though the discussion of some results could be improved.

Clarity: the presentation and organization of the content (method, experiments, related work, etc.) is a very strong point of this contribution. The organization is sensible and predictable, and the presentation is concise and exhaustive, both in text and in figures and tables. The paper is also easily readable to scientific audiences which are not familiar with its specific field of research.

Significance: the presented results are significant and mostly relevant: while FIPER is shown to produce more timely and accurate predictions on the considered environments, scenarios that include humans are not discussed (nor studied empirically, but this might be out of the scope of the paper), even though these are mentioned as highly relevant in the introduction. As a minor point, the discussion of the "sliding window" experiments could also be better motivated.

Originality: the paper presents good but not outstanding originality, which mainly lies in: (1) the idea of combining both OOD detection and action uncertainty to better handle false positives, which in itself seems simple, yet it would be highly impactful to future work; (2) a novel way to use RND, originally developed to solve exploration problems, to detect OOD observations; (3) a new metric to evaluate failure prediction methods.

---

> ### Author Rebuttal · Authors · 2025-07-30
>
> We thank you for your valuable and constructive feedback and sincerely appreciate your recognition of the clarity of our work. We have answered your questions below:
> - **Experiments including humans:** The main motivation behind our work is to increase the interpretability and safety of generative policies. These aspects are important not only in human-centered environments, but especially there. We nevertheless decided against including humans in our real-world and simulated experiments. Our reasoning was as follows: Such experiments are significantly more challenging to conduct in a fair and repeatable manner, as human behavior inevitably exhibits a certain degree of unpredictability. This could only be avoided by simulating non-autonomous humans (e.g., executing fixed pre-recorded motions), but then the sole added value would be the occurrence of a new type of potential failure, namely, collisions with humans. However, rather than collision avoidance, our primary goal is to enable robots to predict more generally if they will fail at their *learned task* due to OOD situations and high action uncertainty, such that policy execution can be stopped, and the robots can ask humans for help. We have emphasized this aspect more clearly in the introduction by rewriting lines 31 to 33 as follows:
>
>    *"In human-centered and safety-critical settings, it is therefore crucial to predict such failures as early as possible during runtime to enable timely intervention [43] or safe fallbacks [8] or to ask human experts to demonstrate the task."*
>
>   In summary, we think that conducting failure prediction experiments involving humans would be a very interesting direction for future research, but we wanted to focus more on improving and evaluating the fundamental capabilities to predict policy failures accurately and early in this work.
>
>   In our opinion, experiments involving humans would not require significant changes to the methodology, but probably a slight adjustment of the metrics. Accuracy and our new timestep-wise-accuracy (TWA) metric (defined in Appendix B.3) weigh true positives and true negatives equally. However, if failures involve breaking expensive items or moving erratically and endangering humans, false negatives (i.e., not detecting failures) are far worse than false positives. For such scenarios, we would propose adding a weighting factor $\\alpha \\geq 0$ to the calculations of balanced accuracy and TWA, for example, changing (41) to
>   $$
>   \\text{BALANCED TWA} = \\frac{1}{1+\\alpha} \\left(\\frac{\\sum_i (1-t_i/T)}{\\#\\mathrm{P}} + \\alpha\\frac{\\#\\mathrm{TN}}{\\#\\mathrm{N}}\\right).
>   $$
>   Then, a small value $\\alpha \\in [0,1]$ could be used if unrecognized failures are more costly than false alarms.
>   If the primary objectives in a human-centered task are to detect all failures and as quickly as possible (even at the cost of many false alarms), our two scores can be combined with a logical OR instead of AND, as shown by the results reported in Table 2. The OR variant achieves lower overall accuracy but can detect 100% of failures. Moreover, the uncertainty score that best satisfies these alternative objectives is our RND-OE score with a small window size $w$ and a small quantile value $1-\\delta$, as can be seen from Figures 9, 10, and 11 in the Appendix. We have added these discussion points to the paper as follows:
>   -  Lines 334 to 337: *"As a variation of FIPER, the use of a logical disjunction (OR) of our two failure predictors provides a powerful alternative when the primary goal is to achieve very low DT and high TPR. This may be particularly desirable if failures could directly endanger people or cause damage to expensive items."*
>   - Lines 985 to 986: *"RND-OE yields a clearer separation between successful and failed rollouts (see Figures 3 and 8) and detects failures more rapidly with both time-varying (Figure 9) and constant (Figure 10) thresholds. This is especially beneficial in scenarios that involve humans, where high sensitivity to different kinds of failures is paramount."*
> - **Moving window discussion**: The tradeoff between detection time and accuracy shown in Figure 4 is an important empirical finding of our work, and we appreciate the reviewer's suggestion to expand the discussion of this tradeoff. The observed higher accuracy of cumulative scores mainly stems from the fact that they rarely raise false alarms, since successful rollouts are usually shorter than failed ones, leaving fewer timesteps to aggregate a large cumulative score that exceeds the threshold. For the same reason, the detection time is also much higher for cumulative scores. In our opinion, it mainly depends on the task whether a decrease in detection time (here: -47\% and -26\%) justifies a drop in accuracy (here: -25\% and -17\%). In domains where rapid response to imminent failures is crucial, such as autonomous surgical assistance or collaborative assembly lines, earlier failure warnings may be highly desirable, even if they mean more false alarms. Conversely, in a logistics scenario, misplacement of objects is often not dangerous, whereas every incorrect failure warning would have to be checked manually by a human and would thus be costly. In this case, a more conservative aggregation over a longer window may better balance operational efficiency and failure prevention. We would like to point out that our moving window method includes the cumulative ($w\\geq \\text{episode length}$) and single-timestep ($w=1$) approaches used in prior works [1, 72] as special cases, and the window size can be chosen based on the characteristics of the task with FIPER. We have added a longer paragraph similar to the above discussion to Appendix C.3 and expanded lines 315 to 317 in the main paper as follows:
>
>   *"In comparison, considering only the most recent timesteps with FIPER enables us to actually predict failures and raise a warning early. This is important, for example, in safety-critical scenarios like surgical assistance or collaborative assembly, even if it causes more false alarms."*
>
> We hope we have addressed your questions and would be happy to elaborate further if needed.

---

> > ### Comment · Reviewer_ZrcY · 2025-08-04
> >
> > Thanks, I appreciate your rebuttal. I do not have further questions and I will maintain my original rating.

---

> > > ### Author Response · Authors · 2025-08-05
> > >
> > > Thank you for considering our rebuttal. We appreciate the time and effort you put into evaluating our work. In your initial comments, you mentioned that if certain concerns were addressed, you would consider raising the significance score and possibly the overall rating. We tried to address those concerns carefully in our response, and we’d be grateful to know if there’s anything further we could clarify or improve that might help you reconsider the score.
> > >
> > > Of course, we fully respect your judgment and gratefully appreciate your feedback either way.

---

### Official Review · Reviewer_Hb4p · 2025-07-03

**Clarity:** 2
**Significance:** 3
**Originality:** 3
**Rating:** 5
**Confidence:** 3

**Summary:**

This paper focuses on the problem of detecting failure for generative imitation learning policies. Their method, FIPER, combines the idea of action and observation failure likelihoods to both consider the policy being in unseen states but also taking suboptimal actions. The insight is that without considering both of these failure modes, the failure detector cannot both let the policy run when it generalizes, but also catch it when it is going to fail. The also enable their method by not requiring access to the training data, and instead they calibrate their method by using a small amount of online data. The contribution of their paper is in the insight that both action and observation information is required to effectively detect failures, and their specific detection method FIPER and an analysis of their method.

**Questions:**

So in general, I think your paper is good. If you could address the concerns I have below mostly with the empirical section I will gladly raise my score as I feel you're approaching an important problem with an interesting method.

So I think you are doing a different sort of uncertainty (at an intent level) but it might be good to mention in section 4.2 that another reason you are doing so is the motivation in this paper: https://arxiv.org/abs/1706.04599 since I think that might be relevant.

I'd appreciate in the related works if for Uncertainty Quantification and OOD Detection you specifically stated how your work slots in with this other work. it seems that you are using some combination of previous OOD Detection methods but which of the works that you state are you building off of?

Small but line 106, why does ReDiffuser not perform optimally? Breaks the flow a bit.

I understand that generative policies produce good results, but is this method specific to generative policies? If no, why limit experiments to generative policies, if yes why so?

I don't think 4.1 is particularly clear. The description of RND makes sense, but 146-154 isn't particularly clear. The questions I have specifically after reading are: It seems that G(.) (the encoder) and g(.) the target network are different, but using a capital G to me implies an instantiation of the target network g. I think it could also benefit from a sentence saying something like "Intuitively, on observations the predictor and target network have seen, they produce similar output, and on novel data, they produce a wide range of results". Maybe this point is obvious but I think it would help clarity.

192 - is this iff a typo or do you mean "if and only if" - If it is the full phrase might want to type it out.

238 implementation/Figure 3 - I found it very easy to miss that you call your metrics RND-OE and ACE before this. Possibly in the subsection headings you should define these. Something like "Detecting OOD Policy Observations - (RND-OE)". I'm sure there are cleaner ways but it confused me at first.

Figure 3 - You should put 95% confidence intervals here, also note the number of trials on the caption. I also found it difficult to quickly determine what all of your labels mean, the Success ID, Success OOD, Fail ID and Fail OOD. I think defining these in the caption especially for figure understanding is important and would improve clarity. Also you sort of say this but making it clear that the most important metric to see here is the gap between Success OOD and Fail ID. Also, the OOD Success and Fail makes sense to me after reading the text but I can't quite figure out what "ID" is in the plots. It would be best if. you can make this clearer.

276 - Probably not important information for main text.

Table 1 - Should have confidence intervals an all these numbers. What does underlined numbers mean? Parens means a poor discrimination but I don't understand the underline.

Figure 4 - This plot can probably be improved. First there are no confidence intervals so it is very difficult to tell if your method has statistical significance. As well, what are all the y axis metrics? Accuracy makes sense, but what is detection time? Percent of episode? From what I can tell here, ACE and FIPER seem to be pretty similar.

From Figure 4, without confidence intervals it seems like your method is basically the same as the ACE component of your method, so what is the full FIPER method really giving you as opposed to ACE?

From Figure 5, without CIs, your method seems almost exactly the same as logpZO. What would RND-OE look like in the Figure 4 comparison? Since it is doing pretty well in this task but would it completely fail with the other window sizes? I'm just a little confused why there is two separate plots here that seem to have similar data. I'd appreciate in the text a clarification about what the difference is.

**Ethical Concerns:**

["NO or VERY MINOR ethics concerns only"]

**Final Justification:**

Authors in their rebuttal addressed my concerns with the empirical section of their paper. I am updating my score to reflect these improvements as they were the main area of issue for the paper.

**Limitations:**

yes

**Quality:**

2

**Strengths And Weaknesses:**

Overall feedback: I think this is an interesting research direction. The method proposed is FIPER which uses a dual metric (both input and action dependent) to determine when a failure is going to happen with limited online data. There result seems to perform well but it is difficult to say without confidence intervals if their method is truly significantly better. The general description of their method makes sense with some comments I provide below. I think that the empirical section isn't super clear, and could use improvements. If the authors are able to make that section more clear (see specific feedback) and it shows their method is strong that would push this paper into an accept since it is an interesting idea and approach, but right now it is a little difficult to tell good (if any) their result truly is.

High level feedback:

Quality: The quality of this paper seems to be reasonable for a NeurIPS paper. I think that in general, the authors have an interesting problem, and describe it decently. Their analysis seems appropriate for the paper, they touch on interesting baselines and provide an interesting ablation. The biggest weakness here is in the empirical section. I don't believe they provide the statistical information to truly asses their results. As well, their descriptions of the tests they run are a bit unclear. I'll provide more feedback below that I think could raise the quality of the empirical section below.

Clarity: The clarity of this paper is ok. I did struggle to understand sections, specifically in the method description, and it took me a few passes to grasp what the authors meant. I'll also provide feedback on this below. I think in general the writing quality was reasonable, I just think parts of the text could use some refining.

Significance: I think the significance of this work is good. From what I can tell, their novelty of using both types of failure detectors in one method is a novel and interesting insight. I believe this is a strength of the work.

Originality: This specific subfield is not my expertise but it seems as if the originality of this paper is good and similar to the significance.

---

> ### Author Rebuttal · Authors · 2025-07-31
>
> Thank you very much for your detailed and constructive feedback and for appreciating the relevance of our work. We first answer your main technical questions:
> - **Restriction to generative policies:** In principle, our method directly applies to any stochastic policy that can sample actions (from which we can compute the action-based ACE score) conditioned on high-dimensional inputs (observation-based RND-OE score). However, the design of our ACE score is motivated by two characteristics that are particularly prevalent in modern generative policies: 1) multimodal action distributions and 2) the use of action chunking, i.e., generating a sequence of actions together to encourage temporal consistency. Both 1) and 2) make it harder to detect high uncertainty from the conditional action distribution $p(A|O)$ compared to, for example, a conventional behavior cloning policy that samples a single-timestep action from a Gaussian with learned mean and variance. Since generative policy formulations have demonstrated superior performance in recent studies (e.g., X. Jia et al., "X-IL: Exploring the Design Space of Imitation Learning Policies", 2025), we have focused on this type of policy in our experiments and used the two most expressive formulations, diffusion and flow matching.
> - **Figure 3**: We agree that this figure would benefit from confidence intervals. Due to limited space, we originally did not include these in the main paper, but provided the distributions of the uncertainty scores, including the 90\% and 10\% quantiles, in Figure 8 in Appendix C.1. These more detailed results support the finding that our two scores excel at distinguishing Success OOD from Fail ID rollouts, and we have added this data to Figure 3, using some of the additional space for the final manuscript.
>
>   We have also followed your suggestion to add the number of seeds (5) and an explanation of the four categories to the caption:
>
>   *"We group the rollouts into four categories along two axes: Success vs. Fail and in-distribution (ID) vs. out-of-distribution (OOD). Robust failure prediction requires distinguishing Success OOD from Fail ID. Our proposed scores (1) and (4) can best make this distinction."*
>
> - **Table 1**: Since our ACE score and the baselines PCA-kmeans and STAC are deterministic, their performance is independent of the random seed. However, we understand that reporting confidence intervals for the other methods would strengthen our empirical evaluation. We have updated Tables 1 and 6 accordingly and summarize the overall results with confidence intervals for five seeds in the following table:
> | Metric       | PCA-kmeans    | logpZO         | RND-A         | STAC          | RND-OE (ours)        | ACE (ours)       | FIPER (ours)        |
> |--------------|---------------|----------------|---------------|---------------|---------------|---------------|----------------|
> | TWA ↑        | 0.57 ± 0      | 0.60 ± 0.016   | 0.56 ± 0.023  | 0.56 ± 0      | 0.58 ± 0.037  | 0.61 ± 0      | **0.62 ± 0.010** |
> | Acc. ↑       | 0.62 ± 0      | 0.69 ± 0.021   | 0.67 ± 0.038  | 0.64 ± 0      | 0.65 ± 0.045  | 0.73 ± 0      | **0.75 ± 0.010** |
> | DT ↓         | (0.11 ± 0)      | 0.34 ± 0.021   | 0.46 ± 0.032  | 0.41 ± 0      | **0.17 ± 0.018** | 0.45 ± 0      | 0.29 ± 0.015   |
>
>   Here, we put detection time (DT) values in brackets if the method discriminates poorly between successes and failures (TPR or TNR below 0.4), similar to the paper. The results underscore that FIPER consistently achieves higher accuracy than the baselines. The low DT of FIPER is only surpassed by its own submodule RND-OE, which is very sensitive to failures but achieves lower accuracy when used alone.
>
>   Besides, underlined numbers in Tables 1 and 6 indicate the second best value in each row, and we added a sentence to the captions to make this clearer.
> - **Figures 4 and 5:** Their main purpose is to compare our approach of aggregating uncertainty over a moving window against a) accumulating scores over all previous timesteps [1] (Figure 4) and b) considering only the current timestep [72] (Figure 5). In these comparisons, we include the respective failure prediction score baseline (action-based STAC [1] for Figure 4 and observation-based logpZO [72] for Figure 5), FIPER, and the submodule of FIPER most similar to the respective baseline (ACE for Figure 4 and RND-OE for Figure 5). Moreover, to stay as close as possible to the setup of these two related works, we use a constant threshold in Figure 4 and a time-varying threshold in Figure 5. Thus, there is no overlap between the data in the two figures, but we understand that this was not entirely clear from the captions. To improve clarity, we have changed the colors in Figure 5 to be different from those in Figure 4, highlighted the threshold type in the captions using an italic font, and expanded the description of the two figures in lines 306 to 322.
>
>   As you suggested, we have also added confidence intervals to Figures 4 and 5. Similar to the table included above, FIPER again performs very consistently across random seeds (within $\\pm 0.015$ for each metric). Thus, the main messages of the figures, i.e., the benefits (fast and robust failure prediction) of aggregating uncertainty over a moving time window, remain unchanged. Importantly, Figures 4 and 5 are not intended as a comparison of different failure prediction scores (e.g., logpZO vs. FIPER). Instead, the main baseline comparison is provided in Table 1 (and an extended version in Table 6). There, we report each method with the best-performing window size $w$ to avoid unfair cherry-picking of $w$ that might favor one particular method.
>
>   As an additional note, the y-axis metrics in Figures 4 and 5 refer to timestep-wise accuracy (TWA), our novel metric for assessing failure prediction performance with a single score, accuracy, and detection time. We normalize detection times by the maximum episode length in all our experiments. Since this information, as well as the meaning of TWA, is hard to spot in the "Metrics" paragraph, we have added those details to all relevant figure captions.
>
> We have also addressed the other issues you raised and made additional changes to the paper to strengthen the discussion of related works and further improve clarity:
> - **ReDiffuser:** This method is designed to increase the reliability of decisions (future trajectories) sampled from a diffusion model. ReDiffuser is tailored to state-based policies and cannot take high-dimensional visual inputs into account, in contrast to our method. The baseline RND-A in our experiments is closely adapted from ReDiffuser, and our results (see, e.g., the table above) show that FIPER substantially outperforms RND-A in terms of accuracy and detection time. As you suggested, we have added a sentence in line 107 to improve the text flow:
>
>   *"However, ReDiffuser is tailored to state-based policies and not designed to take high-dimensional visual inputs into account."*
>
> - **Additional reference**: We agree that the motivation in the mentioned paper (Guo et al., "On Calibration of Modern Neural Networks", 2017) is highly relevant to our work. We have added the reference to the discussion of related work and included a sentence after line 165 in Section 4.2:
>
>   *"To improve model interpretability, which is especially important in decision-making systems [here], we aim to capture uncertainty at an intent level."*
>
> - **Related Work:** We agree with you that emphasizing more how our method relates to other work would add clarity. Our proposed approach comprises two modules, the observation-based RND-OE score and the action chunk-based ACE score. RND-OE builds on random network distillation (RND) [9], which has previously been used to incentivize exploration in reinforcement learning [9], quantify uncertainty [12], and to estimate the reliability of sampled trajectories [21]. The main novelty compared to prior work is that we adopt RND in the *embedding* space of *vision-based* policies and with the specific goal of predicting policy failures. The ACE score is a novel concept, but it has some connections to Bayesian methods [19] and the idea that sampling a batch of actions instead of a single one can be used to detect failures [1]. However, STAC [1] is very different from our approach, as it compares marginal action distributions at consecutive timesteps. The differences between ACE and STAC are also visualized in Figure 2, and we find that ACE performs superior in our experiments, especially for high action multimodality (lines 296 to 299).
>
>   To make these connections to related works clearer, we have added the following after line 93 in the "Uncertainty Quantification and OOD Detection'' paragraph of the related work section:
>
>   *"Our method combines RND in the observation embedding space with a novel entropy-based score, which conceptually has some connections to the idea that sampling a batch instead of a single action can be used to detect failures [1], as well as to Bayesian methods [19]."*
>
> - We have improved Section 4.1 by denoting the encoder by $\\boldsymbol{h}(\\cdot)$ instead of $\\boldsymbol{G}(\\cdot)$ to avoid confusion with the target network $\\boldsymbol{g}(\\cdot)$ and by adding the following below equation (1):
>
>   *"Intuitively, on observations the predictor and target networks have seen, they produce similar outputs; on novel data, however, their outputs differ."*
>
> - We typed out "iff" as "if and only if" in line 192.
> - We added *"(RND-OE)"* and *"(ACE)"* to the headings of Sections 4.1 and 4.2, respectively, to improve readability of the figures and tables that contain these abbreviations.
> - We removed the comment on compute resources in line 276 from the main text, as these details are also provided in Appendix B.5.
>
> We hope to have addressed all your questions and concerns. Please let us know if any further clarification is needed.

---

> > ### Comment · Reviewer_Hb4p · 2025-08-03
> >
> > - Restriction to generative policies: This makes sense I appreciate your clarification.
> >
> > - Figure 3: I see your plot in figure 8 and that is a good one for the appendix. I support you keeping the format of your old plot and just adding in 95% confidence intervals, then leaving the more detailed statistical plot in the appendix. Certainly do it how you would like, and unfortunately you can't show me an updated plot, but I think just adding in 95% CIs would be enough for this plot and would keep it clean. Also thank you for adding clarification I think that helps.
> >
> > - Table 1: I appreciate the updated table and information. This is much improved thank you.
> >
> > - Figures 4 and 5: Your clarifications here are much appreciated. This is also what I was looking for.
> >
> > - The further clarifications also are improved thank you for that.
> >
> > A tiny comment, you also have an "iff" 323. It might be a good idea to switch this one too.
> >
> > I think you have addressed my concerns. I appreciate you making these changes to your work. I'll raise my score.

---

> > > ### Author Response · Authors · 2025-08-04
> > >
> > > Thank you for your comments and for raising your score. We understand your point regarding Figure 3 and have only added 95% CIs there, leaving Figure 8 in the Appendix. We have also replaced the other "iff", thank you for bringing this to our attention.

---

### Author Response · Authors · 2025-08-08

Dear Reviewers Hb4p, Hshp, ZrcY, and a998,

With the discussion period coming to an end, we want to thank you once more for your insightful comments and for the time you dedicated to reviewing our paper. We have done our best to address the concerns raised in our revised manuscript, and we believe that your feedback has significantly improved the quality and clarity of our work.

Best regards,

The authors of submission 24343

---

### Note · Authors · 2025-08-13

Dear AC and Reviewers,

We sincerely thank you for the time and effort you have dedicated to reviewing our work. Below, we summarize the key discussion points and the corresponding changes we made:
- **Clarity and Method Description (Hb4p, a998):**
  - We improved the notation in Section 4.1 and the explanation of RND-OE, and we added multiple references to the methodology details and theoretical results in the Appendix.
  - We expanded the discussion of related work to position our method more clearly within the OOD detection and uncertainty quantification literature.
- **Empirical Evaluation (Hb4p, ZrcY):**
  - We added confidence intervals to all main figures and tables, highlighting the substantial performance improvements achieved by our method.
  - We clarified the purposes of Figures 4 and 5, distinguished their data, and emphasized the benefits of moving windows over alternative score aggregation strategies.
 - **Human-Centered Scenarios (ZrcY):** We expanded the introduction and discussion with specific considerations and tradeoffs in human-centered environments and suggested a slight metric modification for such cases.
- **Combination of Failure Indicators (Hshp, a998):**
  - We clarified the rationale for combining OOD observation detection with action entropy calculation via logical AND and highlighted that this approach can detect 87% of all failures. Additionally, we emphasized that the OR combination can be highly beneficial for safety-critical scenarios as it can predict 100% of failures, albeit at the cost of more false alarms.
  - We described the weighted combination of the detector outputs we tested as an ablation.
- **Task and Action Space Generality (Hshp):** We clarified the computational efficiency of our ACE score and highlighted that our method is applicable to various tasks and action spaces.
- **Hyperparameter and Policy Choices (a998):** We clarified and added a reference to our detailed $\delta$-sensitivity analysis and clarified our task-agnostic choice of policy architectures.
- **Limitations and Future Work (Hshp, Hb4p):** We highlighted the sensitivity to the threshold definition and added remarks on the potential applicability to RL policies.

The manuscript now includes a **clearer methodology presentation, stronger empirical evidence, and an expanded discussion of FIPER's applicability and limitations**. We believe these changes address all concerns raised by the reviewers and significantly strengthen our paper.

---

### Decision · Program_Chairs · 2025-09-17

**Decision:**

Accept (poster)

**Comment:**

This paper studies the problem of failure prediction when executing an imitation learning policy. The key idea is to synergistically consider both sequences of out-of-distribution observations and sustained high action uncertainty. Sufficient empirical experiments are provided to support the effectiveness of the proposed method. The reviewers agree that the studied problem is important, the proposed method is novel, and the empirical improvements are significant. I, therefore, recommend accept.